# A bidirectional switch in the Shank3 phosphorylation state biases synapses toward up- or downscaling

Chi-Hong Wu[1†], Vedakumar Tatavarty[1†], Pierre M Jean Beltran[2†], Andrea A Guerrero[1], Hasmik Keshishian[2], Karsten Krug[2], Melanie A MacMullan[2], Li Li[3], Steven A Carr[2], Jeffrey R Cottrell[3], Gina G Turrigiano[1]*

[1]Department of Biology, Brandeis University, Waltham, United States; [2]Proteomics Platform, Broad Institute of MIT and Harvard, Cambridge, United States; [3]Stanley Center for Psychiatric Research, Broad Institute of MIT and Harvard, Cambridge, United States

*For correspondence: turrigiano@brandeis.edu

†These authors contributed equally to this work

Competing interest: The authors declare that no competing interests exist.

**Abstract** Homeostatic synaptic plasticity requires widespread remodeling of synaptic signaling and scaffolding networks, but the role of post-translational modifications in this process has not been systematically studied. Using deep-scale quantitative analysis of the phosphoproteome in mouse neocortical neurons, we found widespread and temporally complex changes during synaptic scaling up and down. We observed 424 bidirectionally modulated phosphosites that were strongly enriched for synapse-associated proteins, including S1539 in the autism spectrum disorder-associated synaptic scaffold protein Shank3. Using a parallel proteomic analysis performed on Shank3 isolated from rat neocortical neurons by immunoaffinity, we identified two sites that were persistently hypophosphorylated during scaling up and transiently hyperphosphorylated during scaling down: one (rat S1615) that corresponded to S1539 in mouse, and a second highly conserved site, rat S1586. The phosphorylation status of these sites modified the synaptic localization of Shank3 during scaling protocols, and dephosphorylation of these sites via PP2A activity was essential for the maintenance of synaptic scaling up. Finally, phosphomimetic mutations at these sites prevented scaling up but not down, while phosphodeficient mutations prevented scaling down but not up. These mutations did not impact baseline synaptic strength, indicating that they gate, rather than drive, the induction of synaptic scaling. Thus, an activity-dependent switch between hypo- and hyperphosphorylation at S1586 and S1615 of Shank3 enables scaling up or down, respectively. Collectively, our data show that activity-dependent phosphoproteome dynamics are important for the functional reconfiguration of synaptic scaffolds and can bias synapses toward upward or downward homeostatic plasticity.

## Editor's evaluation

The authors survey phosphorylation sites in a large array of proteins to identify targets involved in homeostatic synaptic plasticity in rodent neurons. They identified SHANK3, a critical scaffold protein in the postsynapse, as a target and show that SHANK3 can be phosphorylated and dephosphorylated to regulate homeostatic synaptic plasticity in both directions.

## Introduction

Synaptic scaling is an important form of homeostatic plasticity that bidirectionally adjusts synaptic weights in response to prolonged perturbations in firing, in the correct direction to stabilize neuron

and circuit activity (*Turrigiano, 2008*; *Turrigiano et al., 1998*; *Turrigiano and Nelson, 2004*). Synaptic scaling is expressed through changes in the postsynaptic accumulation of glutamate receptors that then lead to increases or decreases in postsynaptic strength. This form of plasticity is triggered by changes in calcium influx, transcription, and translation (*Dörrbaum et al., 2020*; *Ibata et al., 2008*; *Mao et al., 2018*; *Schanzenbächer et al., 2016*; *Schanzenbächer et al., 2018*; *Schaukowitch et al., 2017*; *Steinmetz et al., 2016*) and involves a complex remodeling of the postsynaptic density that relies on a number of scaffolding and trafficking pathways within individual neurons (*Gainey et al., 2015*; *Hu et al., 2010*; *Louros et al., 2018*; *Steinmetz et al., 2016*; *Sun and Turrigiano, 2011*; *Venkatesan et al., 2020*). While synaptic scaling protocols are known to induce complex changes in the phosphoproteome (*Desch et al., 2021*), the causal roles these changes might play in homeostatic plasticity are largely unexplored. Here, we show that synaptic scaling is accompanied by widespread and dynamic changes in the phosphoproteome that are especially enriched in synapse-associated proteins, and find that activity-dependent bidirectional changes in the phosphorylation state of the synaptic scaffold protein Shank3 are essential for the induction of synaptic scaling up and down.

Shank3 is a multidomain scaffold protein highly enriched at the postsynaptic density (*Naisbitt et al., 1999*) and interacts with a number of other scaffold and signaling proteins known to be important for synaptic scaling, such as Homer1 and the MAGUKs (*Grabrucker et al., 2011*; *Jiang and Ehlers, 2013*). Loss of function of human Shank3 is associated with autism spectrum disorders (ASDs), Phelan–McDermid syndrome, and intellectual disability (*Betancur and Buxbaum, 2013*), indicating that Shank3 plays essential roles within the central nervous system. We recently showed that a cell-autonomous reduction in Shank3 is sufficient to completely abolish synaptic scaling up (*Tatavarty et al., 2020*), but how exactly Shank3 facilitates activity-dependent homeostatic changes in synaptic glutamate receptor abundance, and whether changes in phosphorylation state are important in this process, are unknown.

Here, we employed liquid chromatography, tandem mass spectrometry (LC-MS/MS) to quantitatively profile changes in the synaptic phosphoproteome induced by scaling. We found widespread and temporally complex changes at many phosphosites (>2000 for scaling up and >3000 for scaling down), with strong enrichment for cytoskeletal and synapse-associated proteins. Of these phosphosites, 424 (representing 332 distinct proteins) were bidirectionally regulated during scaling up and down, including Shank3. Further analysis revealed two highly conserved adjacent sites on Shank3 (rat S1615 and rat S1586) that were persistently dephosphorylated during scaling up and were transiently hyperphosphorylated during scaling down. Dephosphorylation of Shank3 during scaling up was maintained by protein phosphatase 2A (PP2A) phosphatase activity, and reversing this dephosphorylation through PP2A inhibition also reversed synaptic scaling up. Finally, we found that mutating S1615 and S1586 to mimic (DD) phosphorylation blocked synaptic scaling up but not down, while mutating these sites to prevent (AA) phosphorylation blocked scaling down but not up. Taken together, these data show that hypophosphorylation of Shank3 through a PP2A-dependent process is essential for maintaining increased postsynaptic strength during scaling up and suggest that the phosphorylation state of Shank3 can bias synapses toward upward or downward synaptic scaling.

## Results
### Synaptic scaling protocols induce widespread and dynamic changes in the phosphoproteome

While changes in phosphorylation of synaptic proteins such as glutamate receptors are thought to play a role in the expression of homeostatic plasticity (*Diering and Huganir, 2018*; *Fernandes and Carvalho, 2016*), the full range of activity-dependent phosphorylation changes induced by synaptic scaling paradigms have not been characterized. We designed an LC-MS/MS experiment to identify dynamic changes in the phosphoproteome during the prolonged increases and decreases in activity that drive homeostatic plasticity. Cultured mouse neocortical neurons were treated in biological duplicates with either tetrodotoxin (TTX, to block action potential firing) or bicuculline (BIC, to enhance firing) for 5 min, 1 hr, 7 hr, or 24 hr, in addition to a control untreated group. Following proteolytic digestion with trypsin and LysC, the peptides were labeled with isobaric mass tag reagents (Tandem Mass Tag TMT) to enable sample multiplexing and precise relative quantification. After mixing, samples were fractionated offline and enriched for phosphopeptides prior to online LC-MS/MS to increase the

depth of coverage of the proteome and phosphoproteome (*Figure 1—figure supplement 1A*). A total of 31,840 phosphosites and 9643 proteins were quantified in the TTX-treated experiment, and 32,635 phosphosites and 9512 proteins were quantified in the BIC-treated experiment, altogether showing the high quality of this proteomics resource. Although biological replicates showed significant variance, the principal component analysis showed that replicate samples clustered together and followed the temporal progression of the experimental design (*Figure 1—figure supplement 1B and C*).

Differential abundance analysis was performed using a moderated *F*-test, identifying proteins and phosphosites with a statistically significant response to BIC or TTX treatment throughout the time course (false discovery rate [FDR]-adjusted p-value<0.10) (*Figure 1—source data 1A–D*). No statistically significant changes in protein abundance were observed during TTX treatment, while 27 proteins were significantly regulated during BIC treatment (*Figure 1—figure supplement 1D*, *Figure 1—source data 1A and C*). These included transcriptional regulators (e.g., Junb, Jund, and Fos) that increased at later time points (7 and 24 hr), suggesting long-term regulation of gene expression programs. In contrast to these modest changes in protein levels, the phosphoproteome data revealed widespread changes in phosphorylation, with 2259 and 3457 phosphosites regulated in response to TTX and BIC treatments, respectively (*Figure 1A and B*, *Figure 1—source data 1B and D*).

These changes in the phosphoproteome were complex and dynamic (*Figure 1A–C*). Cluster analysis identified populations of phosphosites that increased or decreased during TTX treatment, with a range of temporal profiles; the BIC cluster analysis revealed similar complexity (*Figure 1A–C*, top). To understand the biological processes regulated by these temporal profiles, we performed pathway enrichment analysis followed by network integration to summarize pathway enrichment results (*Figure 1C*, bottom, *Figure 1—source data 2*). Neurogenesis pathways were enriched in proteins hyperphosphorylated and hypophosphorylated in response to TTX treatment (*Figure 1C*, neurogenesis group), while neuron projection morphogenesis functions (involved in axon and dendrite formation and dynamics) were enriched in proteins hypophosphorylated in response to BIC (*Figure 1C*, neuron projection morphogenesis group). Numerous pathways associated with synaptic signaling and the synaptic membrane were enriched in clusters that showed increases or decreases during both TTX and BIC treatments (*Figure 1C*, synaptic signaling group). Cytoskeletal pathways were enriched only in phosphosites that increased in response to TTX (cluster 3) and decreased in response to BIC (cluster 4), suggesting that synaptic upscaling involves hyperphosphorylation of cytoskeletal components, and vice versa (*Figure 1C*, cytoskeleton organization group). Finally, splicing and RNA-related processes were enriched in phosphosites with an early and strong hyperphosphorylation response to BIC (*Figure 1C*, splicing and RNA processing group), indicating gene expression regulation via phosphorylation during downscaling. We also observed enrichment of several developmental and morphogenesis pathways in BIC and TTX treatments. To facilitate visualization and exploration of regulated phosphoproteins and pathways in our dataset, we have made available web applications for browsing the TTX (https://proteomics.broadapps.org/HSP_TTX/) and BIC (https://proteomics.broadapps.org/HSP_Bic/) proteomics data.

Synaptic scaling is a bidirectional process, with some overlap in the signaling pathways that regulate up- and downscaling (*Fernandes and Carvalho, 2016*). We noticed a large overlap of synapse-associated pathways across the BIC and TTX clusters. To gain more insight into the processes underlying synaptic scaling, we identified the 424 phosphosites that were bidirectionally regulated by TTX and BIC treatment when considering the average fold change across all time points (*Figure 1D*, *Figure 1—source data 1E*). The majority of these (335) were downregulated by TTX and upregulated by BIC, with a smaller number upregulated by TTX and downregulated by BIC (89). We hypothesized that these phosphosites could be prioritized for mechanistic characterization as they may contribute to the bidirectional processes that underlie homeostatic changes in synaptic strength. These bidirectionally regulated phosphoproteins include the synaptic scaffold proteins Dbn1, Dlgap1, Dlgap4, Homer2, Shank2, and Shank3; a number of neurotransmitter receptors and auxiliary proteins, including Grin3A, Grm5, Gabra2, Gabra5, and Shisa9; and several kinases important in cytoskeletal and synaptic plasticity, including Camk1d and Camk2b (*Figure 1—source data 1E*). A number of motor, trafficking, and sorting proteins were also identified, including the adaptor protein complex AP3 Beta2 subunit (Ap3b2); enhanced expression of the AP3mu subunit was previously identified as an important factor in the routing of AMPAR to the synaptic membrane during scaling up (*Steinmetz*

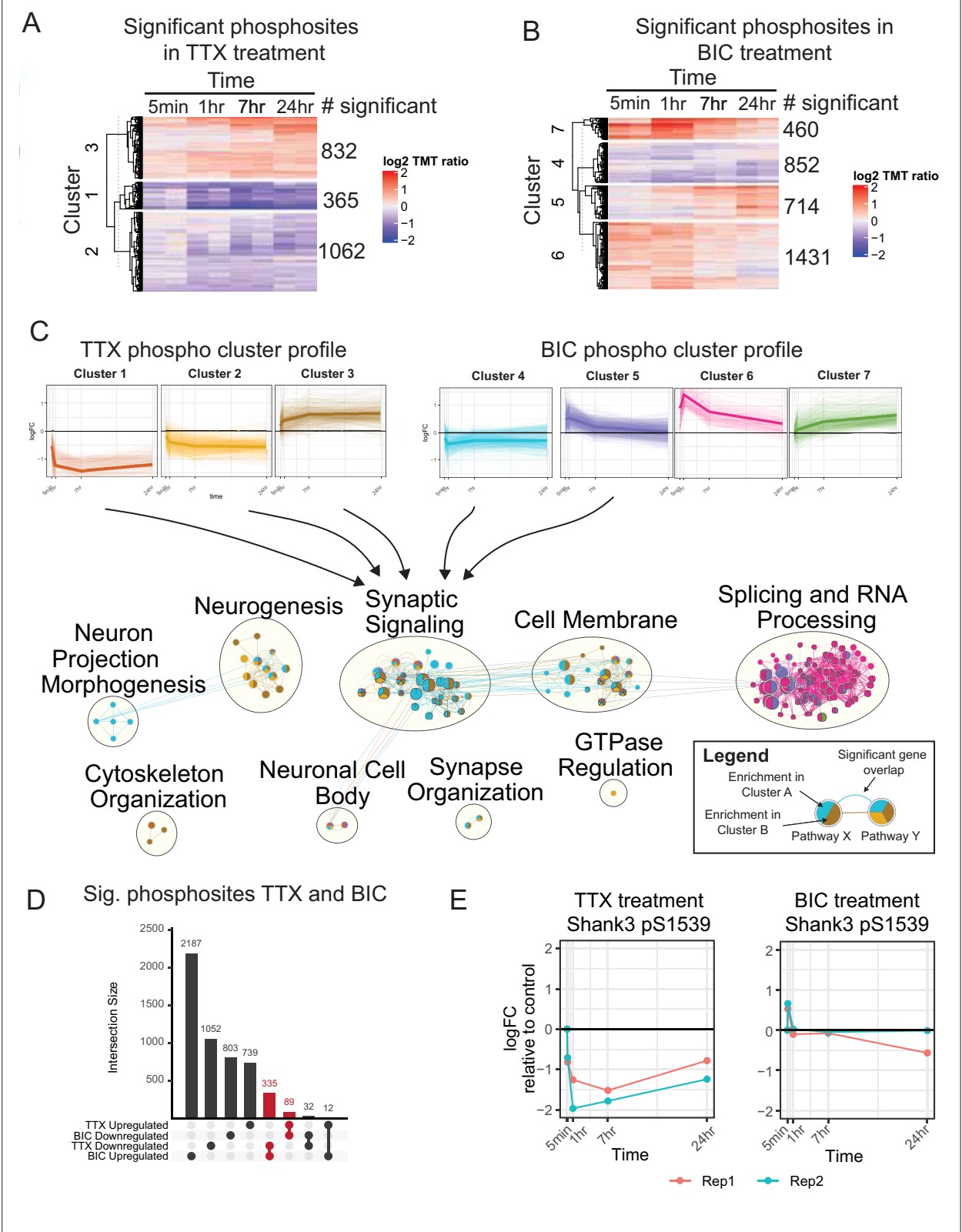

**Figure 1.** The temporal phosphoproteome response induced by synaptic scaling protocols. (**A**) Heatmap showing the abundance (log2 TMT ratios to control) of phosphosites with a significant response to tetrodotoxin (TTX) treatment (*F*-test, adj. p<0.1). Hierarchical clustering shows three major clusters with different temporal profiles. The number of significant phosphosites in each cluster is indicated. (**B**) Heatmap showing the abundance (log2 TMT ratios to control) of phosphosites with a significant response to bicuculline (BIC) treatment (*F*-test, adj. p<0.1). Hierarchical clustering shows

*Figure 1 continued*

four major clusters with different temporal profiles. The number of significant phosphosites in each cluster is indicated. (**C**) Network representation of pathway enrichment results from phosphosites showing a significant response to TTX or BIC treatment. (Top) Temporal profiles are shown for the three TTX clusters and four BIC clusters; thin lines represent individual phosphosites, while thick lines represent the cluster mean. (Bottom) Network showing pathways significantly enriched in each of the TTX/BIC clusters. Pathways are shown as nodes with colors of each node indicating the associated cluster or clusters. Edges connecting the nodes indicate a significant gene overlap between pathways (Jaccard Index > 0.5). Related pathways are clustered and the overall function is summarized in text. The synaptic signaling, neurogenesis, cell membrane, neuronal cell body, and synapse organization clusters show enrichment in response to TTX and BIC treatment in both upregulated and downregulated phosphosites. (**D**) Upset plot showing the number of unique and overlapping regulated phosphosites across BIC and TTX datasets with directionality. Groups highlighted in red represent a total of 424 phosphosites showing regulation in both BIC and TTX datasets, with opposite directionality. (**E**) Temporal profile of mouse Shank3 S1539 phosphosite abundance in response to TTX (left) and BIC (right), displaying opposite response to these treatments; different colors represent two biological replicates. Also see *Figure 1—figure supplement 1*, *Figure 1—source data 1*, and *Figure 1—source data 2*.

The online version of this article includes the following source data and figure supplement(s) for figure 1:

**Source data 1.** Source mass spectrometry (MS) data for *Figure 1*.

**Source data 2.** Source data for pathway enrichment analysis in *Figure 1*.

**Figure supplement 1.** Proteome and phosphoproteome dynamics in synaptic signaling.

*et al., 2016*). Finally, a number of presynaptic proteins and ion channels were bidirectionally regulated during scaling (*Figure 1—source data 1E*). While several of these regulated phosphoproteins have been previously implicated in synaptic scaling, one candidate in particular stood out to us: S1539 of Shank3 – an autism-associated synaptic scaffold protein known to be essential for synaptic scaling up – exhibited bidirectional changes in phosphorylation with different temporal dynamics (*Figure 1E*). These phosphorylation changes occurred without significant changes in protein abundance (*Figure 1—source data 1A and C*). We therefore prioritized Shank3 for in-depth characterization.

## Shank3 is bidirectionally phosphorylated in cultured rat neocortical neurons

Our phosphoproteome screen identified Shank3 as a major synaptic scaffold protein that undergoes robust and bidirectional changes in phosphorylation during synaptic scaling protocols. Shank3 is essential for synaptic scaling up (*Tatavarty et al., 2020*), but it is unknown how Shank3 mediates synaptic plasticity, leading us to wonder whether activity-dependent changes in Shank3 phosphorylation might be critical drivers of synaptic scaling.

To address this question, we first examined whether these activity-dependent phosphorylation changes are conserved across two species (rats and mice) known to express robust synaptic scaling. We isolated Shank3 protein from neurons cultured from postnatal rat visual cortex and analyzed by LC-MS/MS after 24 hr TTX treatment or untreated controls (*Figure 2A*). We found an almost sixfold reduction in phosphorylation at residue rat S1615 (*Figure 2B*). Sequence alignment analysis using Clustal Omega (*Goujon et al., 2010*; *McWilliam et al., 2013*; *Sievers et al., 2011*) showed that Shank3 is highly conserved, with rat Shank3 99.25% identical to its mouse homolog, and human Shank3 94.62% and 95.66% identical to its rat and mouse homologs, respectively. Notably, the sequence around phosphosite S1615 is identical in rat and mouse, with S1615 corresponding to S1539 in mouse Shank3 (*Figure 2C*, bottom). Thus, TTX induces hypophosphorylation at this conserved site in both species. We identified a second residue, rat S1586, that was also hypophosphorylated by TTX treatment, albeit to a lesser extent (*Figure 2B*). Of note, S1586 and S1615 reside in the linker region between the proline-rich and the sterile-alpha-motif (SAM) domains of Shank3 (*Figure 2C*, top).

Given that synaptic scaling is a process that unfolds over a time scale of many hours, we carefully explored the temporal dynamics of Shank3 phosphorylation. In mouse neocortical cultures, there was robust hypophosphorylation of S1539 that was rapidly expressed (within 10 min) and persisted for up to 24 hr during continued TTX application (*Figure 1E*). In contrast, BIC treatment induced a transient hyperphosphorylation at S1539 that was evident after 10 min but reversed within an hour (*Figure 1E*). A similar pattern was observed when Shank3 phosphorylation was assessed in rat cultures using an antibody against phosphorylated S1615 (pS1615, *Figure 2—figure supplement 1*). Activity blockade with TTX induced Shank3 hypophosphorylation at 10 min and 24 hr, while raising activity with picrotoxin (PTX, similar to BIC) induced a transient hyperphosphorylation that reversed at 24 hr (*Figure 2D–G*). Taken together with the mouse data (*Figure 1E*), these data reveal activity-dependent, bidirectional

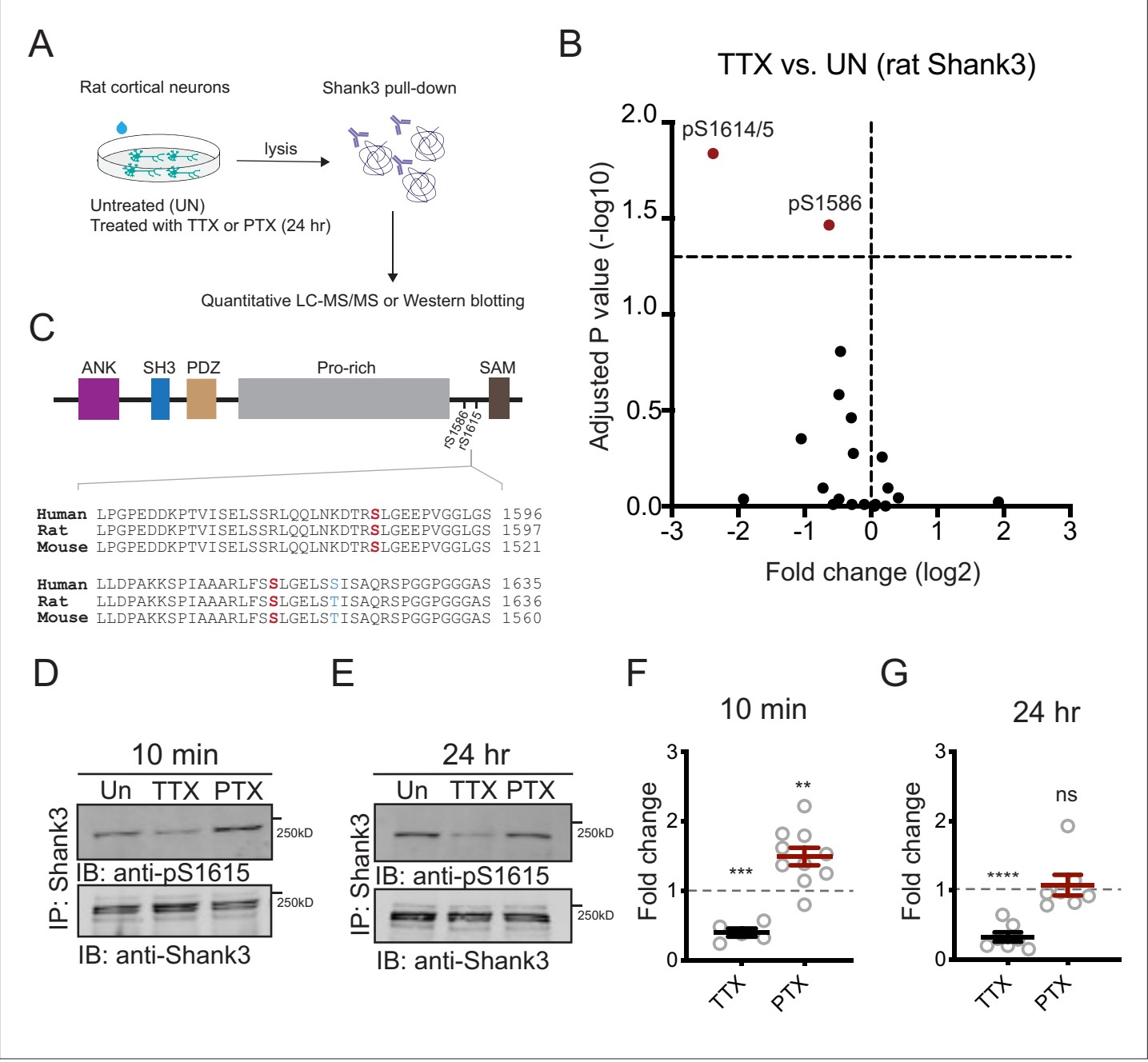

**Figure 2.** Neuronal activity bidirectionally modulates the phosphorylation state of Shank3. (**A**) The experiment protocol for extraction of Shank3 from rat cultured neocortical neurons for further quantitative mass spectrometry (MS) or Western blot analyses. (**B**) Volcano plot of quantitative MS data showing Shank3 residues that were differentially phosphorylated in tetrodotoxin (TTX)-treated samples compared to untreated controls. The log2 values of fold changes, if below zero, indicated hypophosphorylation (paired *t*-test: S1586, adjusted p=0.034142, S1614/5, 0.014444). (**C**) Top: diagram showing the location of S1586 and S1615 within the rat Shank3 protein. Functional domains: ANK = ankyrin repeat; SH3 = SRC homology 3; PDZ = PSD-95/Disc Large/ZO-1; Pro-rich = proline rich; SAM = sterile alpha motif. Bottom: homology comparison of sequences flanking rat S1586 and S1615 (matching mouse S1539) across species (human Shank3: NP_001358973.1; rat Shank3: NP_067708.2; mouse Shank3: UniprotKB: Q4ACU6.3). Phosphosites of interest are labeled in red; the only residue not conserved is shown in blue. (**D, E**) Representative Western blot using an antibody specific for phosphorylated S1615, showing changes in Shank3 phosphorylation after 10 min (**D**) or 24 hr (**E**) treatment with TTX or picrotoxin (PTX). (**F**) Quantification of the fold change of Shank3 S1615 phosphorylation in (**D**). Dashed line indicates the baseline untreated control (one-sample *t*-test: TTX, ***p=0.0005, PTX, **p=0.0035, n = 5 and 10 biological replicates, respectively). (**G**) Quantification of the fold change of Shank3 S1615 phosphorylation in (**E**) (one-sample *t*-test: TTX, ****p<0.0001, PTX, p = 0.6336, n = 7 and 7 biological replicates, respectively). Solid colored horizontal

*Figure 2 continued on next page*

*Figure 2 continued*

lines indicate the mean, and error bars represent SEM. Also see *Figure 2—figure supplement 1*, *Figure 2—source data 1*, and *Figure 2—source data 2*.

The online version of this article includes the following source data and figure supplement(s) for figure 2:

**Source data 1.** Source data for *Figure 2*.

**Source data 2.** Source blot images for *Figure 2*.

**Figure supplement 1.** Validation of the pS1615 antibody.

**Figure supplement 1—source data 1.** Source blot images for *Figure 2—figure supplement 1*.

modifications in the phosphorylation state of Shank3 and show that the temporal dynamics of these phosphorylation changes are conserved across species and culture conditions.

## Phosphorylation state modulates homeostatic changes in the synaptic enrichment of Shank3

Shank3 is highly enriched at the postsynaptic density, where it interacts with a number of synaptic scaffolding and signaling proteins that are important mediators of synaptic scaling (*Gainey et al., 2015*; *Grabrucker et al., 2011*; *Hu et al., 2010*; *Jiang and Ehlers, 2013*; *Shin et al., 2012*; *Sun and Turrigiano, 2011*). We asked whether synaptic scaling protocols might regulate the synaptic abundance of Shank3 by altering its phosphorylation state. Rat visual cortical cultures (used for all subsequent experiments) were treated with TTX or PTX for 24 hr, then fixed and labeled using antibodies against Shank3 and a surface epitope of the postsynaptic AMPA-type glutamate receptor GluA2 (sGluA2, *Figure 3A*), or the presynaptic glutamate transporter VGluT1 (*Figure 3D*). Sites where Shank3 was colocalized with these postsynaptic (sGluA2) or presynaptic (VGluT1) markers of excitatory synapses were identified, and the density of these sites along apical-like pyramidal neuron dendrites, as well as the intensity of the signals at these colocalized sites, was determined. Synaptic scaling protocols induced bidirectional changes in the synaptic accumulation of sGluA2, as expected (*Figure 3A and B*, *Gainey et al., 2015*;; *Ibata et al., 2008*; *Tatavarty et al., 2013*); interestingly, activity blockade with TTX also increased, while enhancing activity with PTX reduced, the synaptic intensity of Shank3 (*Figure 3A and C*). The bidirectional changes in Shank3 synaptic abundance are consistent with the previously reported changes induced by longer scaling protocols (48 hr, *Ehlers, 2003*) and occur in tandem with the changes in AMPAR abundance that underlie synaptic scaling.

To evaluate whether the phosphorylation state of Shank3 influences its synaptic abundance, we generated expression constructs for Shank3 with double point mutations at residues S1586 and S1615, designed to mimic (DD mutants, where serine, S, was replaced with aspartic acid, D) or prevent (AA, where serine was replaced with alanine) phosphorylation at these sites (*Figure 3D*). We then expressed GFP-tagged wild-type, DD, or AA Shank3 at low efficiency in rat cultures, immunolabeled against VGluT1, and quantified the synaptic intensity of the GFP signal in pyramidal neuron apical-like dendrites. The density of colocalized puncta was not different between wild-type Shank3 and Shank3 mutants (*Figure 3F*). In contrast, there was a significant reduction in the accumulation of Shank3 DD at these colocalized sites relative to wild-type Shank3, while the AA mutant was slightly but not significantly increased (*Figure 3E*). These results show that the phosphomimetic Shank3 mutant accumulates less efficiently at synaptic sites and suggests that TTX-induced hypophosphorylation may contribute to the enhanced synaptic abundance of Shank3 during scaling up.

## Increased PP2A activity maintains TTX-induced Shank3 hypophosphorylation

Neuronal activity could alter Shank3 phosphorylation by modulating kinase and/or phosphatase activity (*Figure 4A*). Many serine/threonine kinases could potentially phosphorylate Shank3 at S1615; PKA and CAMKII are two likely candidates due to their activity-dependent activation (*Abel and Nguyen, 2008*; *Shonesy et al., 2014*), role in many forms of synaptic plasticity (*Esteban et al., 2003*; *Shonesy et al., 2014*), and known role in phosphorylating Shank3 (*Perfitt et al., 2020a*; *Wang et al., 2020b*). Pharmacological inhibition of CAMKII (using KN62 or KN93) or PKA (using H89) did not reduce baseline Shank3 phosphorylation. However, inhibition of either kinase prevented the transient Shank3 hyperphosphorylation induced by PTX (*Figure 4B and C*).

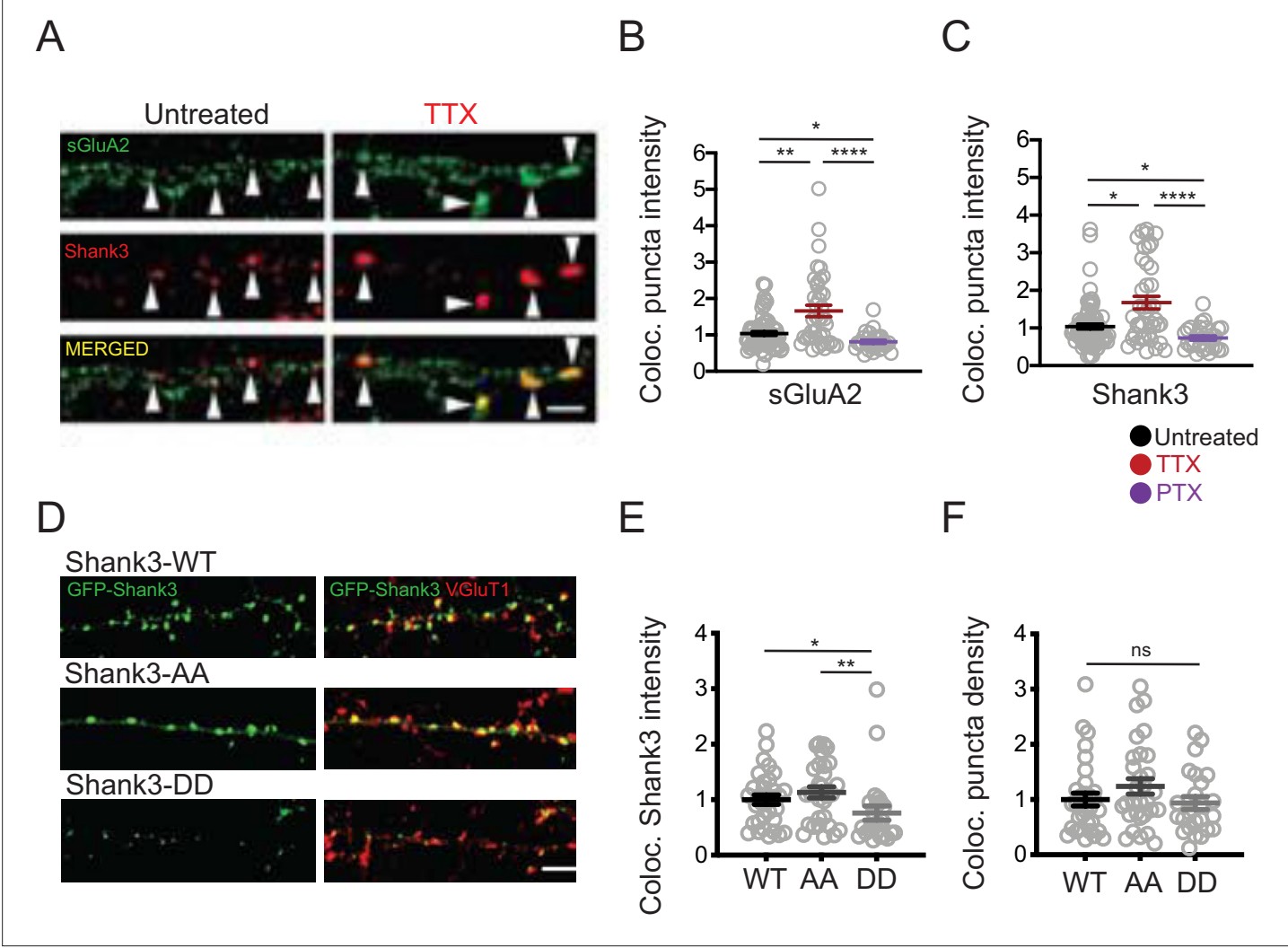

**Figure 3.** Phosphorylation state modulates activity-dependent changes in the synaptic enrichment of Shank3. (**A**) Representative images of synaptic puncta colocalized with surface GluA2 (sGluA2) and Shank3 in neuron dendrites ± tetrodotoxin (TTX) (scale bar = 5 μm). (**B**) Quantification of synaptic sGluA2 intensity changes induced by scaling up and down protocols (number of neurons: untreated, n = 77, TTX, n = 40, picrotoxin [PTX], n = 29; Kruskal–Wallis test with post-hoc Dunn's multiple comparison tests: Un vs. TTX, **p=0.0034, Un vs. PTX, *p=0.0408, TTX vs. PTX, ****p<0.0001). (**C**) Quantification of synaptic Shank3 intensity during scaling up and down protocols (Kruskal–Wallis test with post-hoc Dunn's tests: Un vs. TTX, *p=0.0155, Un vs. PTX, *p=0.0205, TTX vs. PTX, ****p<0.0001). (**D**) Representative images of synaptic localization of wild-type Shank3 and Shank3 phospho-mutants (scale bar = 5 μm). (**E**) Quantification of synaptic intensity of Shank3 phospho-mutants (number of neurons: WT, n = 33, AA, n = 30, DD, n = 24; Kruskal–Wallis test with post-hoc Dunn's tests: WT vs. AA, p>0.9999, WT vs. DD, *p=0.0395, AA vs. DD, **p=0.0039). (**F**) Quantification of the density of synaptic puncta containing Shank3 phospho-mutants (number of neurons: WT, n = 32, AA, n = 30, DD, n = 24; Kruskal–Wallis test: p=0.2814). For imaging experiments here and below, each data point represents a single pyramidal neuron, and data were collected from at least four independent experiments. Also see *Figure 3—source data 1*.

The online version of this article includes the following source data for figure 3:

**Source data 1.** Source data for *Figure 3*.

Since scaling up is accompanied by robust and persistent hypophosphorylation of Shank3, we then turned our attention to the potential role of activity-dependent phosphatase activity in this hypophosphorylation. Protein phosphatase 2A (PP2A) plays an important role in some forms of synaptic plasticity (*Colbran, 2004*; *Launey et al., 2004*; *Mauna et al., 2011*; *Winder and Sweatt, 2001*; *Woolfrey, 2015*); therefore, it seemed a likely candidate to dephosphorylate Shank3. We immunoprecipitated the catalytic subunit of PP2A (PP2Ac) from cells treated with TTX and subjected it to an in vitro PP2A activity assay to detect any changes in the activity of PP2A during scaling up. When compared

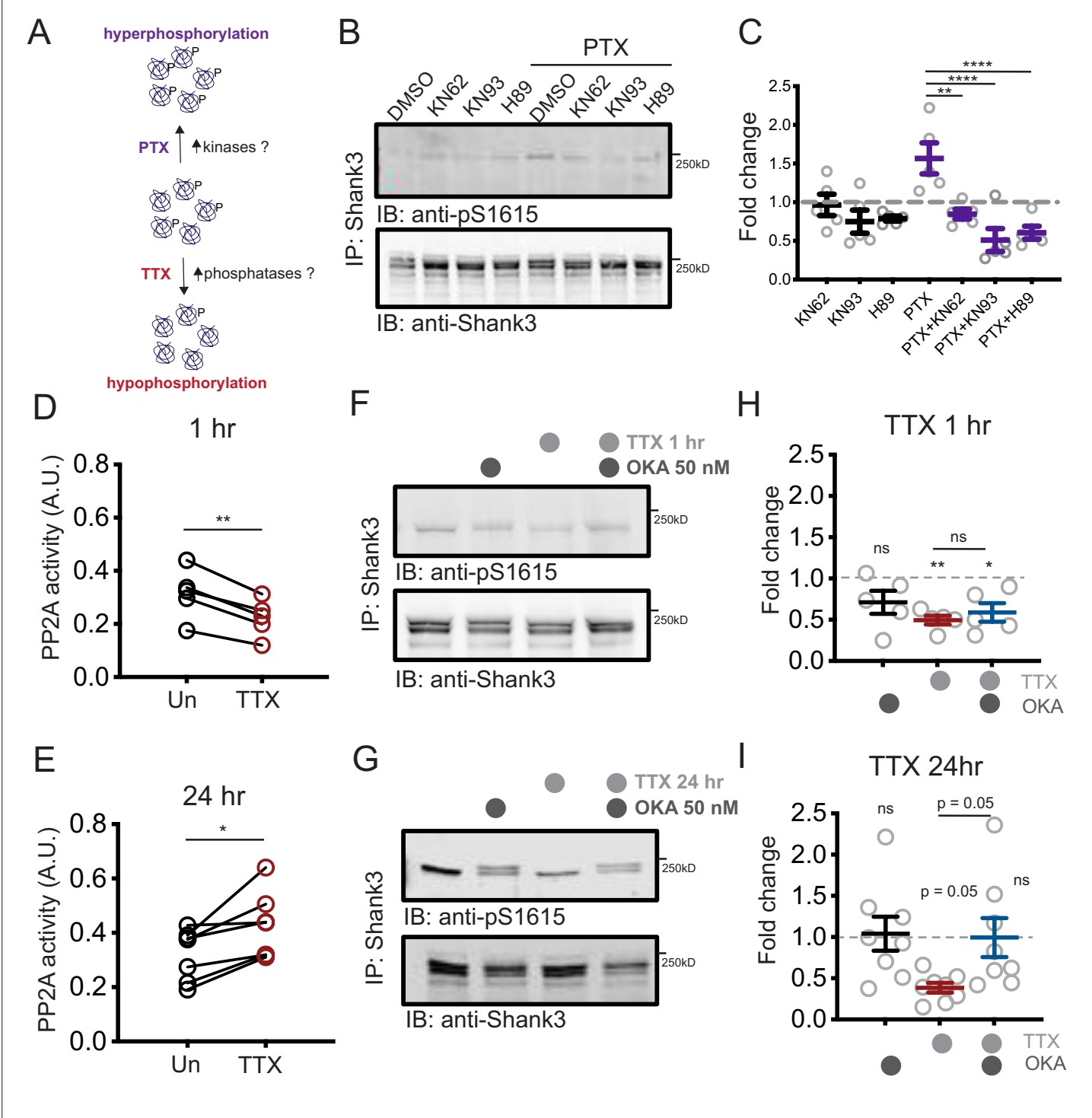

**Figure 4.** Increased PP2A activity maintains tetrodotoxin (TTX)-induced Shank3 hypophosphorylation. (**A**) Diagram showing the potential roles of kinases and phosphatases in regulating activity-dependent Shank3 phosphorylation. (**B**) Representative Western blot showing the impacts of inhibiting CAMKII (KN62, KN93) or PKA (H89) on Shank3 phosphorylation at baseline and upon TTX treatment. (**C**) Quantification of S1615 phosphorylation in (**B**) (two-way ANOVA with post-hoc Tukey's test: DMSO vs. KN62, p>0.9999, DMSO vs. KN93, p=0.8148, DMSO vs. H89, p=0.9112, DMSO vs. picrotoxin (PTX), *p=0.0406, PTX vs. PTX/KN62, **p=0.0040, PTX vs. PTX/KN93, ****p<0.0001, PTX vs. PTX/H89, ****p<0.0001, n = 5 biological replicates). Dashed line indicates the DMSO control. (**D**) Quantification of PP2A activity after 1 hr TTX treatment (Un, n = 5, TTX, n = 5; paired *t*-test: **p=0.0018). (**E**) Quantification of PP2A activity after 24 hr TTX treatment (Un, n = 7, TTX, n = 7; paired *t*-test: *p=0.0129). (**F, G**) Western blot analyses showing changes in S1615 phosphorylation after 1 hr (**F**) or 24 hr (**G**) TTX treatment, with inhibition of PP2A by okadaic acid (OKA, 50 nM) during the

*Figure 4 continued on next page*

Figure 4 continued

last hour of treatment. (**H**) Quantification of S1615 phosphorylation in (**F**) (two-way ANOVA with post-hoc Tukey's test: Un vs. OKA, 0.1723, Un vs. TTX, **p=0.0076, Un vs. TTX/OKA, *p=0.0311, TTX vs. TTX/OKA, p=0.8942, n = 5 biological replicates). Dashed line indicates the baseline untreated control. (**I**) Quantification of S1615 phosphorylation in (**G**) (two-way ANOVA test with post-hoc Tukey's test: Un vs. OKA, = 0.9979, Un vs. TTX, p=0.0503, Un vs. TTX/OKA, p>0.9999, TTX vs. TTX/OKA, p=0.0531, n = 8 biological replicates). Also see *Figure 4—figure supplement 1*, *Figure 4—source data 1*, and *Figure 4—source data 2*.

The online version of this article includes the following source data and figure supplement(s) for figure 4:

**Source data 1.** Source data for *Figure 4*.

**Source data 2.** Source blot images for *Figure 4*.

**Figure supplement 1.** PP1 regulates baseline phosphorylation of Shank3.

**Figure supplement 1—source data 1.** Source data for *Figure 4—figure supplement 1*.

**Figure supplement 1—source data 2.** Source blot images for *Figure 4—figure supplement 1*.

to untreated samples, we observed an initial reduction in PP2A activity after 1 hr of TTX treatment (*Figure 4D*), which then shifted to an increase after 24 hr treatment (*Figure 4E*).

These biphasic changes in phosphatase activity led us to hypothesize that PP2A may have a critical role in maintaining, rather than inducing, Shank3 hypophosphorylation during the late phase of scaling up. To test this idea, we treated cells with TTX for 1 or 24 hr, and during the last hour of the regimen, we applied okadaic acid (OKA), a phosphatase inhibitor that preferentially blocks PP2A activity at a low concentration (50 nM, *Bialojan and Takai, 1988*; *Cohen et al., 1989*; *Ishihara et al., 1989*; *Pribiag and Stellwagen, 2013*). We then immunoprecipitated Shank3 and assessed the phosphorylation state of Shank3. As expected, inhibition of PP2A did not alter Shank3 hypophosphorylation in the early phase of scaling (1 hr TTX, *Figure 4F and H*); in contrast, the hypophosphorylation was reversed to baseline when OKA was introduced during the last hour of a 24 hr treatment with TTX (*Figure 4G and I*). We also observed a second band of phosphorylated Shank3 after OKA treatment, suggesting that there may be other phosphosites regulated by the OKA treatment that collectively induce a shift in molecule weight. OKA treatment alone did not alter the baseline phosphorylation level of Shank3, indicating that the role of PP2A in dephosphorylating Shank3 only manifested during prolonged neuronal inactivity (*Figure 4H and I*). We wondered whether other phosphatases such as protein phosphatase 1 (PP1) could also be involved in Shank3 phosphorylation, so we repeated this experiment using a higher concentration of OKA known to inhibit both PP2A and PP1 (500 nM, *duBell et al., 2002*; *Ishihara et al., 1989*; *Pribiag and Stellwagen, 2013*). This concentration of OKA induced significant hyperphosphorylation of Shank3 under baseline conditions (*Figure 4—figure supplement 1A and B*). Taken together, these data suggest that PP1 controls baseline Shank3 phosphorylation, while PP2A undergoes an activity-dependent biphasic change in activity that maintains the hypophosphorylated state of Shank3 during activity blockade.

## PP2A is required for TTX-induced synaptic enrichment of Shank3

We next assessed whether PP2A activity influences the synaptic enrichment of Shank3 during scaling up using the more specific PP2A inhibitor fostriecin (FST, *Walsh et al., 1997*). Neurons were treated with TTX for 24 hr, and 10 nM FST was added during the final hour. TTX treatment increased both the intensity (*Figure 5A and B*) and density (*Figure 5C*) of the synaptic Shank3 puncta. Inhibition of PP2A prevented the increase in Shank3 puncta density (*Figure 5C*) and reduced (but did not eliminate) the increase in Shank3 intensity at remaining Shank3 puncta (*Figure 5B*). FST treatment alone did not impact the baseline synaptic intensity or density of Shank3 puncta (*Figure 5B and C*), supporting the view that PP2A regulates phosphorylation and synaptic Shank3 localization only during activity blockade. We repeated this experiment with OKA and confirmed that PP2A inhibition diminished synaptic enrichment of Shank3 during activity blockade (*Figure 5—figure supplement 1A and B*).

Because PP1 regulates Shank3 phosphorylation under basal conditions, we also tested whether PP1 inhibition influences synaptic Shank3 clustering by applying tautomycetin (TAUT), a specific PP1 inhibitor (10 nM, *Mitsuhashi et al., 2001*). TAUT treatment alone reduced the baseline intensity of synaptic Shank3 without disrupting the density of synaptic Shank3 puncta (*Figure 5D–F*). Taken together with our direct measurements of Shank3 phosphorylation (*Figure 4F–I*, *Figure 4—figure*

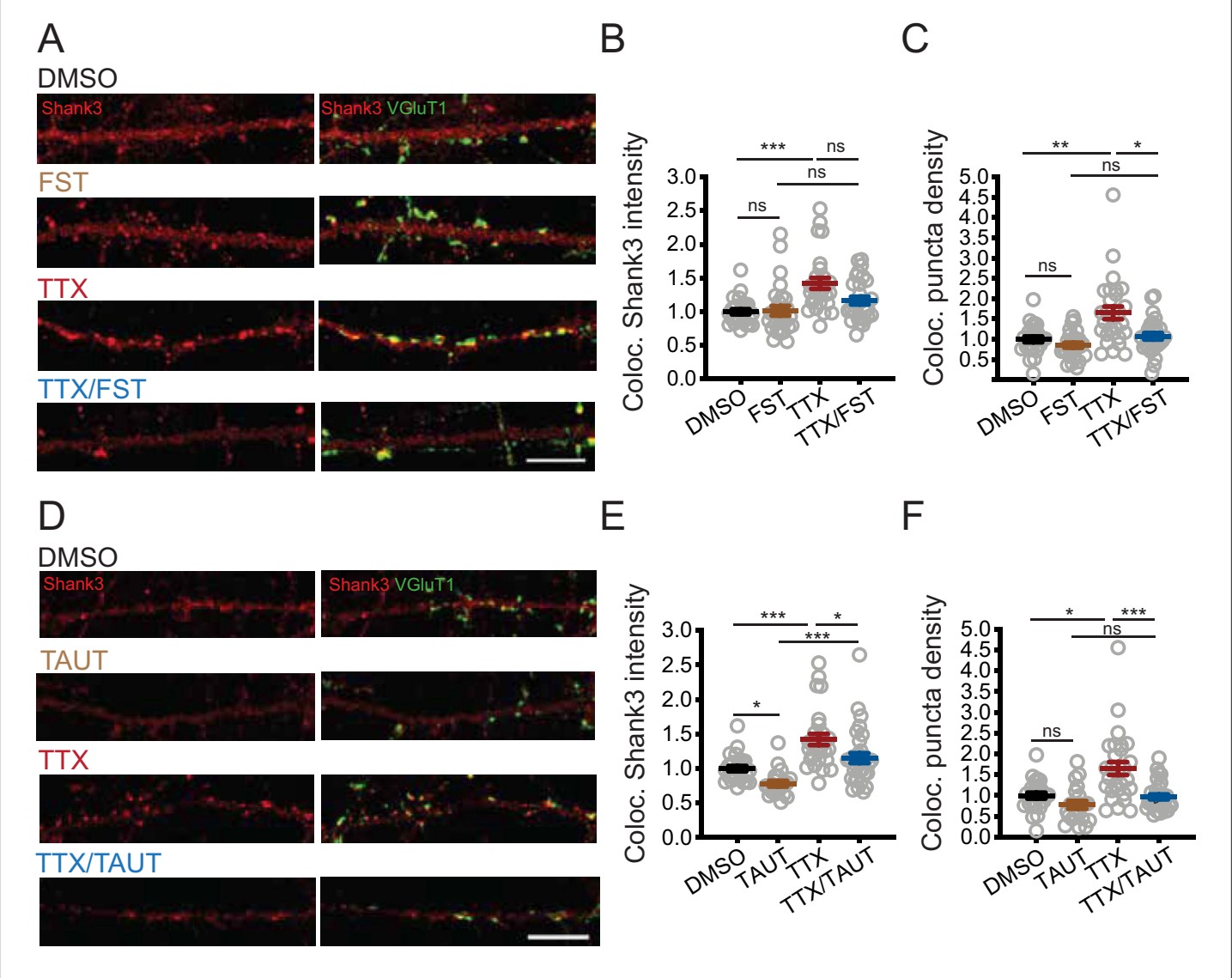

**Figure 5.** PP2A activity is required for tetrodotoxin (TTX)-induced synaptic enrichment of Shank3. (**A**) Representative images of synaptic enrichment of endogenous Shank3 upon treatment with TTX and PP2A inhibitor fostriecin (FST) (scale bar = 10 µm). (**B**) Quantification of synaptic Shank3 intensity in (**A**) (number of neurons: DMSO, n = 26, FST, n = 28, TTX, n = 28, TTX/FST, n = 29; Kruskal–Wallis test with post-hoc Dunn's tests: DMSO vs. FST, p>0.9999, DMSO vs. TTX, ***p=0.0002, FST vs. TTX/FST, p=0.1259, TTX vs. TTX/FST, p=0.1292). (**C**) Quantification of density of synapses containing Shank3 in (**A**) (Kruskal–Wallis test with post-hoc Dunn's tests: DMSO vs. FST, p=0.9458, DMSO vs. TTX, **p=0.0051, FST vs. TTX/FST, p=0.2446, TTX vs. TTX/FST, *p=0.0273). (**D**) Representative images of synaptic enrichment of endogenous Shank3 upon treatment with TTX and PP1 inhibitor tautomycetin (TAUT) (scale bar = 10 µm). (**E**) Quantification of synaptic Shank3 intensity in (**D**) (number of neurons: DMSO, n = 26, TAUT, n = 21, TTX, n = 28, TTX/TAUT, n = 32; Kruskal–Wallis test with post-hoc Dunn's tests: DMSO vs. TAUT, *p=0.0315, DMSO vs. TTX, ***p=0.0006, TAUT vs. TTX/TAUT, ***p=0.0002, TTX vs. TTX/TAUT, *p=0.0392). (**F**) Quantification of density of synapses containing Shank3 in (**D**) (Kruskal–Wallis test with post-hoc Dunn's tests: DMSO vs. TAUT, p=0.2450, DMSO vs. TTX, *p=0.0116, TAUT vs. TTX/TAUT, p=0.6552, TTX vs. TTX/TAUT, ***p=0.0007). Also see *Figure 5—figure supplement 1* and *Figure 5—source data 1*.

The online version of this article includes the following source data and figure supplement(s) for figure 5:

**Source data 1.** Source data for *Figure 5*.

**Figure supplement 1.** Inhibition of PP2A by okadaic acid (OKA) reverses tetrodotoxin (TTX)-induced synaptic enrichment of Shank3.

**Figure supplement 1—source data 1.** Source data for *Figure 5—figure supplement 1*.

*supplement 1*), these data suggest that PP1 influences baseline synaptic Shank3 clustering, while PP2A is recruited during activity deprivation to dephosphorylate Shank3 and promotes its enrichment at synapses.

## The phosphorylation state of Shank3 is critical for enabling bidirectional synaptic scaling

Shank3 is necessary for synaptic scaling up (*Tatavarty et al., 2020*), but whether changes in its phosphorylation state are critical for this (or indeed any other) function of Shank3 is unknown. To test this, we transfected neurons with our phosphomimetic (DD) mutant of Shank3 to determine whether preventing hypophosphorylation would block synaptic scaling up. We performed whole-cell patch-clamp recordings from transfected neurons and measured AMPAR-mediated miniature excitatory postsynaptic currents (mEPSCs), a physiological measure of postsynaptic strength. In neurons over-expressing Shank3 WT, TTX induced the normal increase in the mEPSC amplitude that is the classic measure of scaling up (*Figure 6A and C*). Strikingly, overexpression of the DD mutant completely blocked scaling up (*Figure 6B and D*), suggesting that dephosphorylation of Shank3 at these sites is essential for its induction. Overexpression of DD (*Figure 6A–D*) or AA (*Figure 6E–H*) had no significant effect on baseline mEPSC amplitude.

While a normal complement of Shank3 is essential for scaling up (*Tatavarty et al., 2020*), whether it is also required for scaling down is unknown. To investigate this, we expressed a short-hairpin RNA at low efficiency to reduce synaptic Shank3 by ~50% in transfected pyramidal neurons (*Tatavarty et al., 2020*). Treatment of cultures with PTX for 24 hr induced the expected reduction in mean mEPSC amplitude and a shift in the cumulative probability distribution of amplitudes toward smaller values (*Figure 6—figure supplement 1A and B*), indicative of synaptic scaling down. Knockdown of Shank3 completely prevented scaling down in transfected neurons (*Figure 6—figure supplement 1A and B*), indicating that Shank3 is necessary for both directions of synaptic scaling. Because elevated neuronal activity induces transient hyperphosphorylation of S1615 (*Figure 2D–G*), we next wondered if this hyperphosphorylation was necessary for scaling down. Indeed, although scaling down was intact in neurons expressing WT Shank3, it was absent in neurons expressing the phosphodeficient AA mutant (*Figure 6E–H*).

Our data show that hypophosphorylation of Shank3 is necessary for scaling up, while hyperphosphorylation is necessary for scaling down. This suggests that the phosphorylation state of Shank3 acts as a switch to enable up- or downscaling; if so, scaling up should be intact in the AA (non-phosphorylatable) mutant, and scaling down should be intact in the DD (phosphomimetic) mutant. To test this, we transfected neurons with the AA mutant and treated them with TTX, or the DD mutant and treated with BIC, and then quantified synaptic GluA2 to measure synaptic scaling. We found that scaling up was indeed preserved in the AA mutant; likewise, scaling down was preserved in the DD mutant (*Figure 6—figure supplement 2*); notably, expression of these mutants does not by itself drive scaling up or down as baseline mEPSC amplitude was unaffected by AA or DD expression (*Figure 6*). Thus, phosphorylation of Shank3 at residues S1586 and S1615 blocks scaling up but is permissive for scaling down, while dephosphorylation at these same residues blocks scaling down but is permissive for scaling up.

## PP2A sustains scaling up through hypophosphorylation of Shank3

If the ability of PP2A inhibition to reverse scaling up is dependent upon enabling phosphorylation at S1586 and S1615 (*Figure 4I*), then it should be prevented by expression of the AA mutant, which cannot be phosphorylated at these sites. To test this, we transfected neurons with either WT or AA Shank3, treated cultures with TTX for 24 hr, added FST during the last hour of treatment, and then measured synaptic scaling by quantifying the synaptic accumulation of GluA2. Compared with TTX treatment alone, we found that FST treatment significantly reduced the synaptic intensity of both sGluA2 and Shank3, as well as the density of colocalized puncta, in cells expressing Shank3 WT (*Figure 7A–D*). In contrast, FST treatment was unable to reduce GluA2 and Shank3 accumulation in neurons expressing the AA mutant (*Figure 7A–D*). Taken together, these data demonstrate that PP2A maintains the expression of synaptic scaling up by dephosphorylating Shank3 at S1586 and S1615.

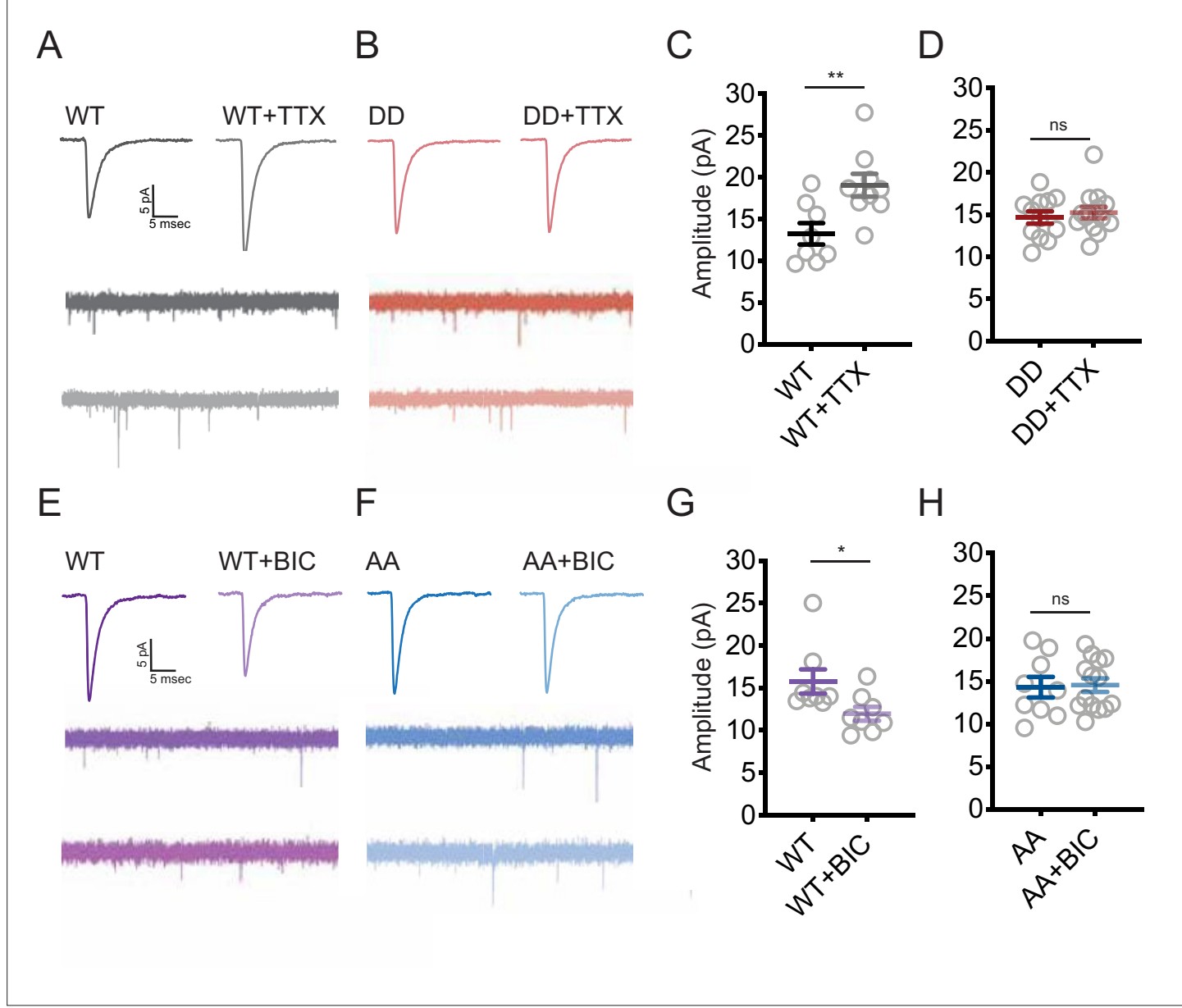

**Figure 6.** Changes in the phosphorylation state of Shank3 are crucial for bidirectional synaptic scaling. (**A, B**) Representative miniature excitatory postsynaptic current (mEPSC) recordings from neurons overexpressing Shank3 WT (**A**) or DD mutant (**B**) during scaling up. (**C**) Quantification of average mEPSC amplitude in (**A**) (WT, n = 8, WT + tetrodotoxin [TTX], n = 9; unpaired two-tailed *t*-test: **p=0.0074). (**D**) Quantification of average mEPSC amplitude in (**B**) (number of neurons: DD, n = 12, DD + TTX, n = 14; unpaired two-tailed *t*-test: p=0.5708). (**E, F**) Representative traces of mEPSCs recorded from neurons overexpressing Shank3 WT (**E**) or AA mutant (**F**) during scaling down. (**G**) Quantification of average mEPSC amplitude in (**E**) (number of neurons: WT, n = 8, WT + bicuculline [BIC], n = 8; Mann–Whitney test: *p=0.0148). (**H**) Quantification of average mEPSC amplitude in (**F**) (AA, n = 9, AA + BIC, n = 14; unpaired two-tailed *t*-test: p=0.8612). Also see *Figure 6—figure supplement 1*, *Figure 6—figure supplement 2*, and *Figure 6—source data 1*.

The online version of this article includes the following source data and figure supplement(s) for figure 6:

**Source data 1.** Source data for *Figure 6*.

**Figure supplement 1.** Shank3 is required for synaptic scaling down.

**Figure supplement 1—source data 1.** Source data for *Figure 6—figure supplement 1*.

**Figure supplement 2.** Scaling up and down remain intact in neurons overexpressing Shank3 AA and DD, respectively.

**Figure supplement 2—source data 1.** Source data for *Figure 6—figure supplement 2*.

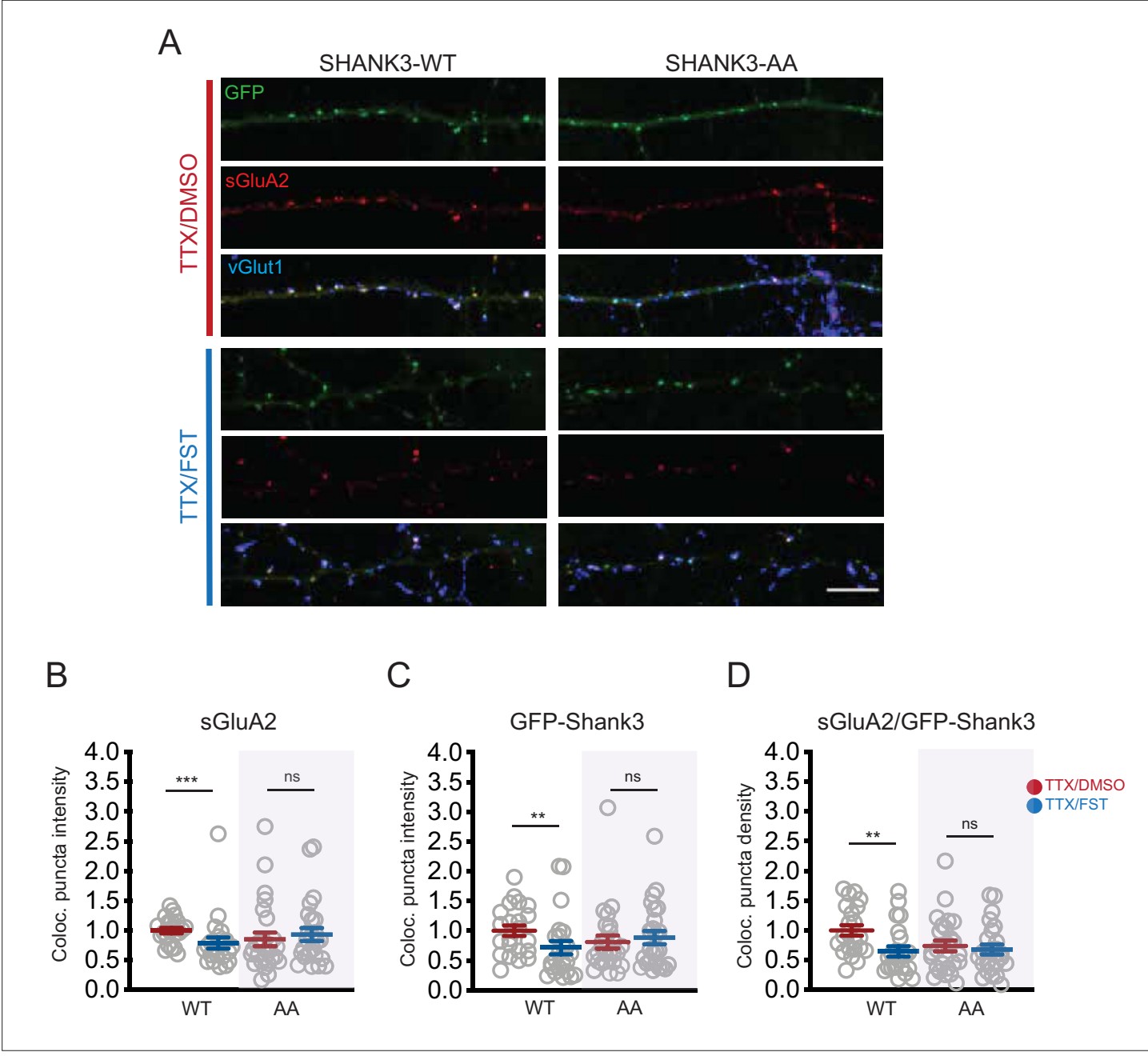

**Figure 7.** Brief PP2A inactivation reverses scaling up. (**A**) Representative images showing the effects of 1 hr fostriecin (FST) treatment on synaptic sGluA2 intensity in neurons expressing Shank3 WT or AA, after 24 hr of tetrodotoxin (TTX) to scale up synaptic strengths (scale bar = 10 μm). (**B**) Quantification of synaptic sGluA2 intensity in (**A**) (number of cells: WT/TTX, n = 22, WT/TTX/FST, n = 23, AA/TTX, n = 26, AA/TTX/FST, n = 25; Mann–Whitney test: WT/TTX vs. WT/TTX/FST, ***p=0.0007, AA/TTX vs. AA/TTX/FST, p=0.3739). (**C**) Quantification of synaptic Shank3 intensity in (**A**) (Mann–Whitney test: WT/TTX vs. WT/TTX/FST, **p=0.0090, AA/TTX vs. AA/TTX/FST, p=0.7296). (**D**) Quantification of the density of puncta containing sGluA2 and Shank3 (Mann–Whitney test: WT/TTX vs. WT/TTX/FST, **p=0.0016, AA/TTX vs. AA/TTX/FST, p=0.7017). Each data point indicates a cell, and the total number (**n**) was pooled from five independent experiments. Also see *Figure 7—source data 1*.

The online version of this article includes the following source data for figure 7:

**Source data 1.** Source data for *Figure 7*.

## Discussion

Homeostatic synaptic scaling is a bidirectional process that modifies the accumulation of synaptic glutamate receptors through a complex remodeling of postsynaptic scaffolding, trafficking, and signaling networks (*Gainey et al., 2015*; *Hu et al., 2010*; *Louros et al., 2018*; *Steinmetz et al., 2016*; *Sun and Turrigiano, 2011*; *Tatavarty et al., 2020*; *Venkatesan et al., 2020*). Notably, scaling up and down have been reported to support the therapeutic effects of ketamine and lithium on depression and bipolar disorder, respectively; thus, a complete understanding of how bidirectional synaptic scaling is regulated could potentially help advance targeted treatments of these mood disorders (*Kavalali and Monteggia, 2020*). While transcriptional and translational regulation of this process has been intensively studied, it remains unclear whether post-translational modifications also play a causal role. Here, we analyzed the temporal dynamics of the phosphoproteome during the prolonged changes in activity that drive synaptic scaling up and down and found widespread and dynamic regulation of phosphorylation that was especially enriched in pathways related to synaptic scaffolding and signaling. We then focused on Shank3, a synaptic scaffold protein known to be essential for synaptic scaling (*Tatavarty et al., 2020*), and which exhibited robust and bidirectional changes in phosphorylation during scaling protocols. We found that Shank3 is dephosphorylated at two sites (S1586 and S1615) during scaling up and hyperphosphorylated during scaling down. These changes in phosphorylation did not impact basal synaptic function, but were necessary for the expression of synaptic scaling. Finally, we found that dephosphorylation of these sites via PP2A activity was essential for the maintenance of synaptic scaling up. These data show that Shank3 undergoes an activity-dependent switch between hypo- and hyperphosphorylation at S1586/ S1615 that is necessary to enable scaling up or down, respectively. More broadly, widespread changes in the phosphoproteome are likely to be instrumental in reconfiguring pre- and postsynaptic scaffold and signaling pathways during homeostatic plasticity.

Mass spectrometry-based proteomics has been successful in elucidating phosphorylation signaling and adaptation mechanisms in various forms of synaptic plasticity (*Guan et al., 2011*; *Hwang et al., 2021*; *Li et al., 2016*). In this work, we began by exploring changes induced by homeostatic scaling protocols using a quantitative proteomic and phosphoproteomic methodology. The multiplexing strategy and extensive peptide fractionation provided deep coverage of proteins and phosphosites while reducing technical variability. We identified over 5200 temporally regulated phosphosites (FDR-adjusted p-value<0.10), with only subtle changes in protein levels: only 27 proteins in the BIC treatment condition showed differential abundance changes with our statistical significance criteria. Dörrbaum et al. previously characterized the proteome synthesis, degradation, and turnover during homeostatic scaling (*Dörrbaum et al., 2020*). Their work identified hundreds of proteins with differential abundance during 7 days of treatment; the longer treatment conditions and higher number of replicates they used may be necessary to capture these relatively modest abundance changes. Nonetheless, our work shows that extensive phosphorylation events with large effect sizes occur at a point in time (24 hr) when there is robust synaptic scaling and relatively small changes in protein abundance. Importantly, 424 of these phosphosites – including Shank3 – exhibited bidirectional changes in phosphorylation during scaling up and down. The global pattern of change in the phosphoproteome during synaptic scaling up and down, and the enrichment in pathways related to synaptic scaffolding and signaling, is consistent with a recent study that used a label-free MS approach (*Desch et al., 2021*). Thus, synaptic scaling results in widespread bidirectional regulation of the phosphoproteome, and these changes may contribute to the regulation of a wide range of cell-biological processes that contribute to the expression of homeostatic plasticity.

In addition to synaptic scaling of excitatory synapses, prolonged changes in activity produce a wide range of homeostatic network adaptations that include changes in intrinsic excitability and inhibition (*Turrigiano, 2011*). Proteomics revealed a number of voltage-gated ion channel subunits that are differentially phosphorylated during scaling up and down protocols that are candidate contributors to these intrinsic excitability changes (*Figure 1—source data 1*). These ion channels include hyperpolarization-activated cyclic nucleotide-gated (HCN) channels (Hcn1 and Hcn2), delayed rectifier and inwardly rectifying potassium channel subunits (Kcnb1 and Kcnj3), and L- and T-type calcium channel subunits (Cacna1c and Cacna1i). The function of phosphorylation at the sites identified on these channel subunits will be an interesting avenue of exploration for understanding the mechanisms of intrinsic homeostatic plasticity. Similarly, we identified changes in phosphorylation in a network of

presynaptic scaffold and release proteins, including Bassoon (Bsn), Piccolo (Pclo), Synapsin 2 and 3 (Syn2 and Syn3), and Synaptotagmin 11 and 17 (Syt11 and Syt17), which may contribute to presynaptic adaptations. Coupled with the widespread changes in postsynaptic scaffolding and signaling proteins (including Dlgap1 and 4, Homer2, Shank2, Shank3, and Shisa9), these data support the notion of coordinated pre- and postsynaptic adaptations during synaptic scaling. Finally, we identified bidirectional changes in phosphorylation of a number of neurotransmitter receptors known to be important for inhibitory and excitatory homeostatic plasticity, including the GABA$_A$R subunits Gabra2 and Gabra5, the NMDAR subunit Grin3A, and the mGluR5 subunit Grm5. In contrast, although bidirectional changes in phosphorylation of GluA2 at Y876 have been reported after 48 hr of activity manipulation (*Yong et al., 2020*), we failed to detect such changes; this may reflect the shorter treatment times used here (24 rather than 48 hr) and underscores the dynamic nature of activity-dependent phosphorylation events. Although we have focused our attention here on sites that undergo bidirectional phosphorylation, it is important to note that the machinery that drives scaling up and down (and possibly other forms of homeostatic plasticity) are not entirely overlapping (for scaling up-specific examples, see *Stellwagen and Malenka, 2006*; *Tan et al., 2015*; for scaling down examples, see *Sun and Turrigiano, 2011*; *Wang et al., 2017*), and so some of the large number of unidirectionally affected phosphosites we identified may also prove to be mechanistically important.

Changes in the phosphorylation state of Shank3 were of particular interest to us since Shank3 is critical for synaptic scaling up (*Tatavarty et al., 2020*), interacts with many synaptic partners also known to be important for homeostatic plasticity (*Gainey et al., 2015*; *Hu et al., 2010*; *Shin et al., 2012*), and is strongly associated with ASDs and intellectual disability (*Betancur and Buxbaum, 2013*). We found that Shank3 is bidirectionally phosphorylated during scaling on two conserved sites in the linker region between the proline-rich and SAM domains in both mouse and rat neocortical neurons and that these phosphorylation changes showed similar dynamics between species, with immediate and prolonged dephosphorylation induced by activity deprivation, and transient early hyperphosphorylation when activity is raised. One of these sites (reported as S1511, corresponding to rat S1586 in our dataset) was also identified in Desch et al.'s report, while rat S1615 was not (*Desch et al., 2021*). This likely reflects more comprehensive coverage of Shank3 in our assays. These changes in phosphorylation state gate changes in the synaptic abundance of Shank3 during scaling and are essential for the expression of synaptic scaling up and down (respectively). The different time courses of Shank3 phosphorylation changes during scaling up and down are intriguing and suggest that the temporal dynamics of the synaptic scaling machinery are more complex than previously appreciated. In particular, while the transient hyperphosphorylation of Shank3 is necessary to initiate scaling down, it need not be maintained for the subsequent slow removal of synaptic AMPAR that underlies the reduction in synaptic strength. In contrast, upscaling requires the continuous and active hypophosphorylation of Shank3 as transiently reversing this hypophosphorylation after 24 hr of scaling by inhibiting PP2A rapidly reverses scaling up.

Activity-dependent changes in Shank3 phosphorylation could be achieved through the altered activity of kinases, phosphatases, or both. Several activity-dependent kinases are known to regulate Shank3: ERK2 and its downstream target ribosome S6 kinase (RSK) phosphorylate Shank3 at S1134/S1163/S1253 (*Wang, 2020a*) and S1648 (*Thomas et al., 2005*), respectively, while CaMKII can target S782 (*Jeong et al., 2021*) and S1586 (*Dosemeci and Jaffe, 2010*). Moreover, both CaMKII and PKA can phosphorylate S685 (*Perfitt et al., 2020a*; *Wang et al., 2020b*). Similar to S685, we found that both CaMKII and PKA regulate hyperphosphorylation of S1615 during heightened neuronal activity; these effects could be direct or indirect. Further, inhibition of CAMKII and PKA did not significantly affect baseline phosphorylation of S1615, suggesting the involvement of additional kinases. Altogether, these data highlight the complexity of the kinase network that regulates Shank3 phosphorylation. In addition to being a target of multiple kinases, Shank3 dephosphorylation is actively maintained by phosphatases: we found that PP1 modulates basal levels of Shank3 phosphorylation, while PP2A undergoes a delayed increase in activity upon TTX treatment that is required to maintain the hypophosphorylated state of Shank3 during late scaling. One model that could explain these temporal dynamics during scaling up is that reduced kinase (such as CAMKII) activity may account for the early stage of Shank3 hypophosphorylation, which is then maintained by the delayed activation of PP2A.

While no pharmacological manipulation is completely specific, our finding that PP2A participates in the activity-dependent regulation of Shank3 phosphorylation is bolstered by (1) the use of two distinct

inhibitors and (2) the lack of effect of these inhibitors on baseline Shank3 phosphorylation. Using a different approach, we also find that PP2A activity is enhanced by prolonged activity blockade. Nonetheless, future experiments using alternative approaches such as endogenous PP2A inhibitors (e.g., inhibitors of protein phosphatase 2A [I2PPA]) (*Tanimukai et al., 2005*) or genetic manipulations will be needed to further corroborate our conclusion. Moreover, given that all of our experiments were conducted using whole-cell lysates, we cannot exclude the possibility that Shank3 phosphorylation might be regulated by molecular factors downstream of PP2A through an indirect mechanism.

How does the phosphorylation state of Shank3 enable scaling up and down? Our observation that phosphorylation at S1586/ S1615 influences synaptic enrichment of Shank3 suggests that this phosphorylation contributes to the recruitment and/or stability of Shank3 at the synapse. This connection between Shank3 phosphorylation and synaptic clustering is supported by the recent observation that dephosphorylation at S782 increases the synaptic enrichment of Shank3 (*Jeong et al., 2021*). S1586 and S1615 are located in the linker region between the proline-rich domain and the SAM domain. The SAM domain promotes synaptic targeting (*Boeckers et al., 2005*), oligomerization, and stabilization of Shank3 (*Baron et al., 2006*; *Hayashi et al., 2009*; *Naisbitt et al., 1999*), suggesting that phosphorylation at these sites might influence synaptic Shank3 enrichment by regulating the function of the SAM domain. The proximity of S1586 and S1615 to the upstream proline-rich domain, which interacts with important cytoskeletal and signaling elements including cortactin and Homer1 (*Naisbitt et al., 1999*), also raises the possibility that the phosphorylation state of these sites modulates the interaction of Shank3 with local signaling pathways essential for synaptic plasticity. It is likely that these two mechanisms are not mutually exclusive; instead, there could be temporal and spatial coordination that facilitates each direction of synaptic scaling. For instance, during scaling up, persistent dephosphorylation of Shank3 may locally stabilize Shank3 in the synapse and modulate its interaction with other synaptic proteins that result in GluA2 accumulation. By contrast, during scaling down, the early but transient hyperphosphorylation may remove Shank3 from the synapse and then initiate molecular programs that sustain scaling down. The latter speculation is supported by reports that Shank3 undergoes activity-dependent translocation from the synapse to the nucleus (*Grabrucker et al., 2014*) and interacts with CAMKII to facilitate phosphorylation of nuclear CREB (*Perfitt et al., 2020b*). More broadly, our data suggest that changes in Shank3 phosphorylation state act as a permissive switch that then enables transcription-dependent trafficking and scaffolding pathways to enhance or reduce AMPAR accumulation at the synapse.

Shank3 is a complex, multiply phosphorylated protein that is highly enriched in the postsynaptic density and interacts both directly and indirectly with cytoskeletal elements, synaptic scaffold proteins, glutamate receptors, and synaptic signaling elements (*Grabrucker et al., 2011*; *Jiang and Ehlers, 2013*). Shank3 is thus perfectly placed to act as a signaling and scaffolding hub that can coordinate the activity of the multiple cell-biological processes to gate the induction of homeostatic changes in synaptic strength. Taken together, our data show that bidirectional and temporally complex changes in Shank3 phosphorylation are necessary for synaptic scaling and suggest that the function of Shank3 within the postsynaptic density is dynamically modulated by its phosphorylation state to switch it from a configuration that enables scaling up to one that enables scaling down. This has important implications for our understanding of bidirectional synaptic plasticity, and how loss of Shank3 contributes to synaptic and circuit dysfunction.

## Materials and methods

All animal procedures were performed according to NIH guidelines and were approved by the Broad Institute of MIT and Harvard IACUC (mouse cultures) or the Brandeis University IACUC (rat cultures).

### Key resources table

| Reagent type (species) or resource | Designation | Source or reference | Identifiers | Additional information |
|---|---|---|---|---|
| Strain, strain background (*Rattus norvegicus*) | Long–Evans | Charles River Laboratories | Strain: 006; RRID:RGD_2308852 | |
| Strain, strain background (*Mus musculus*) | C57BL/6 | Charles River Laboratories | Strain:027; RRID:IMSR_CRL:27 | |

*Continued on next page*

*Continued*

| Reagent type (species) or resource | Designation | Source or reference | Identifiers | Additional information |
|---|---|---|---|---|
| Cell line | HEK293T | ATCC | CRL-3216; RRID:CVCL_0063 | |
| Antibody | Anti-Shank3 (guinea pig polyclonal) | Synaptic Systems | Cat# 162304; RRID:AB_2619863 | IF (1:1000); WB (1:1000) |
| Antibody | Anti-pS1615 (rabbit polyclonal) | This paper | | WB (1:1000) Detailed information can be found in section 'Generation and validation of phosphospecific Shank3 antibody', and the reagent is available upon request |
| Antibody | Anti-HA (rabbit monoclonal, C29F4) | Cell Signaling Technology | Cat# 3724S; RRID:AB_1549585 | IP (1:500) |
| Antibody | Anti-HA (chicken polyclonal) | Aves Labs | Cat# ET-HA100; RRID:AB_2313511 | WB (1:1000) |
| Antibody | Anti-rabbit Alexa Fluor 680 (donkey polyclonal) | Jackson ImmunoResearch | Cat# 711-625-152; RRID:AB_2340627 | WB (1:5000) |
| Antibody | Anti-guinea pig IRDye 800CW (donkey polyclonal) | LI-COR | Cat# 925-32411; RRID:AB_2814905 | WB (1:5000) |
| Antibody | Anti-chicken IRDye 800CW (donkey polyclonal) | LI-COR | Cat# 925-32218; RRID:AB_2814922 | WB(1:5000) |
| Antibody | Anti-GFP (chicken polyclonal) | Aves Labs | Cat# GFP-1020; RRID:AB_2307313 | IF (1:1000) |
| Antibody | Anti-VGluT1 (guinea pig polyclonal) | Synaptic Systems | Cat# 135304; RRID:AB_887878 | IF (1:1000) |
| Antibody | Anti-VGluT1 (rabbit polyclonal) | Synaptic Systems | Cat# 135302; RRID:AB_887877 | IF (1:1000) |
| Antibody | Anti-GluA2 (mouse monoclonal) | Gift from Eric Gouaux, OHSU | | IF (1:1000) |
| Antibody | Anti-chicken Alexa 488 (goat polyclonal) | Thermo Fisher Scientific | Cat# A-11039; RRID:AB_142924 | IF (1:400) |
| Antibody | Anti-mouse Alexa 555 (goat polyclonal) | Thermo Fisher Scientific | Cat# A-21424; RRID:AB_141780 | IF (1:400) |
| Antibody | Anti-guinea pig Alexa 555 (goat polyclonal) | Thermo Fisher Scientific | Cat# A-21435; RRID:AB_2535856 | IF (1:400) |
| Antibody | Anti-guinea pig Alexa 647 (goat polyclonal) | Thermo Fisher Scientific | Cat# A-21450; RRID:AB_2735091 | IF (1:400) |
| Antibody | Anti-rabbit Alexa 647 (goat polyclonal) | Thermo Fisher Scientific | Cat# A-21245; RRID:AB_2535813 | IF (1:400) |
| Recombinant DNA reagent | Shank3 Short Hairpin (PVLTHM) | Gift from Chiara Verpelli | | *Verpelli et al., 2011* |
| Recombinant DNA reagent | pDEST53-CMV-Cycle3GFP-Shank3 WT (short-hairpin insensitive) | Gift from Chiara Verpelli | | *Verpelli et al., 2011* |
| Recombinant DNA reagent | pDEST53-CMV-Cycle3GFP-Shank3 S1586A/ S1615A | This paper | | Detailed information can be found in the section 'Expression constructs and generation of Shank3 mutants,' and the reagent is available upon request |
| Recombinant DNA reagent | pDEST53-CMV-Cycle3GFP-Shank3 S1586D/ S1615D | This paper | | Detailed information can be found in the section 'Expression constructs and generation of Shank3 mutants,' and the reagent is available upon request |
| Recombinant DNA reagent | pDEST53-CMV-HA-Shank3 WT | This paper | | Detailed information can be found in the section 'Expression constructs and generation of Shank3 mutants,' and the reagent is available upon request |
| Recombinant DNA reagent | pDEST53-CMV-HA-Shank3 S1586A/ S1615A | This paper | | Detailed information can be found in the section 'Expression constructs and generation of Shank3 mutants,' and the reagent is available upon request |

*Continued on next page*

*Continued*

| Reagent type (species) or resource | Designation | Source or reference | Identifiers | Additional information |
|---|---|---|---|---|
| Recombinant DNA reagent | pAAV-CMV-PI-EGFP-WPRE-bGH | Gift from James M. Wilson | Addgene# 105530; RRID:Addgene_105530 | |
| Commercial assay or kit | Lipofectamine 2000 | Thermo Fisher Scientific | Cat# 11668-027 | |
| Commercial assay or kit | Gibson Assembly Master Mix | New England Biolabs | Cat# E2611S | |
| Commercial assay or kit | Lambda protein phosphatase | New England Biolabs | Cat# P0753S | |
| Commercial assay or kit | BCA Protein Assay Kit | Thermo Fisher Scientific | Cat# 23227 | |
| Commercial assay or kit | Protein-G Magnetic Beads | Thermo Fisher Scientific | Cat# 88847 | |
| Commercial assay or kit | SimplyBlue SafeStain | Thermo Fisher Scientific | Cat# LC6060 | |
| Commercial assay or kit | PP2A Immunoprecipitation Phosphatase Assay Kit | Millipore | Cat# 17-313 | |
| Commercial assay or kit | Ni-NTA Superflow Agarose Beads | QIAGEN | Cat# 30410 | |
| Chemical compound, drug | Tetrodotoxin | Tocris | Cat# 1069 | |
| Chemical compound, drug | Bicuculline methobromide | Tocris | Cat# 0109 | |
| Chemical compound, drug | Picrotoxin | Sigma-Aldrich | Cat# P1675 | |
| Chemical compound, drug | Okadaic acid | Santa Cruz | Cat# sc-3513 | |
| Chemical compound, drug | Tautomycetin | Tocris | Cat# 2305 | |
| Chemical compound, drug | Fostriecin | Tocris | Cat# 1840 | |
| Chemical compound, drug | KN62 | Tocris | Cat# 1277 | |
| Chemical compound, drug | KN93 | Tocris | Cat# 1278 | |
| Chemical compound, drug | H89 | Tocris | Cat# 2910 | |
| Chemical compound, drug | Sequencing-grade trypsin | Promega | Cat# V5111 | |
| Chemical compound, drug | Tandem Mass Tag (TMT) 10plex | Thermo Fisher Scientific | Cat# 90110 | |
| Software, algorithm | Image Lab Software | Bio-Rad | RRID:SCR_014210 | https://www.bio-rad.com/en-us/product/image-lab-software?ID=KRE6P5E8Z&source_wt=imagelabsoftware_surl |
| Software, algorithm | ZEN Black | Zeiss | RRID:SCR_018163 | https://www.zeiss.com |
| Software, algorithm | Metamorph | Molecular Devices | RRID:SCR_002368 | http://www.moleculardevices.com/Products/Software/Meta-Imaging-Series/MetaMorph.html |
| Software, algorithm | Fiji | Fiji | RRID:SCR_002285 | http://fiji.sc |
| Software, algorithm | GraphPad Prism | GraphPad | RRID:SCR_002798 | http://www.graphpad.com/ |
| Software, algorithm | IGOR pro | Wavemetrics | RRID:SCR_000325 | https://www.wavemetrics.com/products/igorpro/igorpro.htm |
| Software, algorithm | Spectrum mill v.7.00.208 | Agilent Technologies | | |
| Software, algorithm | R v 4.0 | The R Foundation | RRID:SCR_001905 | https://www.R-project.org/ |

*Continued on next page*

*Continued*

| Reagent type (species) or resource | Designation | Source or reference | Identifiers | Additional information |
|---|---|---|---|---|
| Software, algorithm | Cytoscape v 3.8.2 | Cytoscape | RRID:SCR_003032 | http://cytoscape.org |
| Software, algorithm | EnrichmentMap | EnrichmentMap | RRID:SCR_016052 | http://baderlab.org/Software/EnrichmentMap |
| Other | Odyssey Blocking Buffer | LI-COR | Cat# 927-40000 | Detailed information can be found in the section 'Immunoprecipitation and Western blotting' |
| Other | DAPI-Fluoromount-G | SouthernBiotech | Cat# 0100-20 | Detailed information can be found in the section 'Immunocytochemistry' |

## Neuronal cultures, transfections, and drug treatments

### Rat cultures

Timed-pregnant Long–Evans rats were obtained from Charles River. Primary neuronal cultures were dissociated from the visual cortex of newborn pups (postnatal days 1-3) and plated onto glass-bottomed dishes pre-seeded with glial feeders as previously described (*Gainey et al., 2015*; *Tatavarty et al., 2020*). All the experiments were performed from 7 to 10 days in vitro (DIV), during which neurons were sparsely transfected with the following constructs using Lipofectamine 2000 (Thermo Fisher Scientific). To exogenously express Shank3 phospho-mutants, Shank3 constructs (2500 ng per dish) were transfected. To knock down endogenous Shank3, an shRNA targeting Shank3 (*Tatavarty et al., 2020*; *Verpelli et al., 2011*) was used. For better visualization of neurons during recording, an empty GFP vector was co-transfected in both conditions. In the imaging experiments that measured endogenous Shank3, an empty GFP vector (500 ng per dish) was transfected to delineate the neurons. To induce scaling up, 6 hr (*Figure 6—figure supplement 2*) or 24 hr after transfection (the remaining experiments) neurons were treated with TTX (5 µM, Tocris) for ~16 hr. To induce scaling down, PTX (100 µM, Sigma-Aldrich) or BIC (20 µM, Tocris) were used for the same duration. In the experiments where phosphatases were inhibited, OKA (50 or 500 nM, Santa Cruz), FST (10 nM, Tocris), or TAUT (10 nM, Tocris) were introduced during the last hour of the scaling-inducing regimen. Because the phosphatase inhibitors were dissolved in DMSO, the same volume of DMSO was added to matched sister cultures for the same duration as controls. In all imaging and electrophysiology experiments, we only analyzed excitatory pyramid neurons, which were marked by their stereotypical morphology and typically exhibited robust synaptic scaling (*Turrigiano et al., 1998*).

For Western blotting and mass spectrometry, dissociated neurons were plated onto 10 cm plates without the glial feeders and underwent the same treatments described above.

### Mouse cultures

Timed pregnant C57BL/6 mice were acquired from Charles River. Tissue collection was performed at E17. Cortex was dissected in ice-cold Hibernate E medium (Thermo Fisher Scientific) supplemented with 2% B27 supplement (Thermo Fisher Scientific) and 1% Pen/Strep (Thermo Fisher Scientific). Brain tissues were digested in Hibernate E containing 20 U/ml papain, 1 mM L-cysteine, 0.5 mM EDTA (Worthington Biochem kit), and 0.01% DNase (Sigma-Aldrich) for 10 min. Neurons were dissociated and plated at a density of $6 \times 10^6$/dish onto poly-D-lysine coated 100 mm plates (Biocoat, Corning). Cortical neurons were seeded and maintained in NbActiv1 (BrainBits Inc, Springfield, IL) and grown at 37°C in 95% air with 5% $CO_2$ humidified incubator for 16 days. Cortical neurons were left untreated for control or treated with 1 µM TTX or 20 µM BIC for 5 min, 1 hr, 7 hr, and 24 hr before collection for proteomics and phosphoproteomics analysis.

## Cell lines

The HEK293T cell line (CRL-3216) was purchased from the American Type Culture Collection (ATCC), where it was authenticated by STR profiling and tested negative for mycoplasma contamination (Hoechst DNA stain method).

## Proteomic profiling of mouse neuronal cultures

### In-solution digestion

Neuronal cell pellets containing ~$6.6 \times 10^6$ cells were lysed for 30 min at 4°C in urea lysis buffer (8 M urea, 50 mM Tris-HCl pH 8.0, 75 mM NaCl, 1 mM EDTA, 2 µg/µl aprotinin [Sigma-Aldrich], 10 µg/µl leupeptin [Roche], 1 mM phenylmethylsulfonyl fluoride [PMSF] [Sigma-Aldrich], 10 mM NaF, and 1:100 phosphatase inhibitor cocktails 2 and 3 [Sigma-Aldrich]) and cleared by centrifugation at 20,000 × g. Samples were reduced with 5 mM dithiothreitol (DTT) for 1 hr at 25°C, followed by alkylation with 10 mM iodoacetamide for 45 min at 25°C. Samples were diluted with 50 mM Tris-HCl pH 8.0 to a final urea concentration of 2 M preceding enzymatic digestion. Proteins were digested with endo-proteinase LysC (Wako Laboratories) for 2 hr at 25°C followed by overnight digest with sequencing-grade trypsin (Promega) at 25°C (enzyme-to-substrate ratios of 1:50). Following digestion, samples were acidified to a concentration of 1% formic acid (FA) and cleared by centrifugation at 20,000 rcf. Remaining soluble peptides were desalted using a reverse-phase tC18 SepPak cartridge (Waters). Cartridges were conditioned with 1 ml 100% acetonitrile (MeCN) and 1 ml 50% MeCN/0.1% FA, then equilibrated with 4 × 1 ml 0.1% trifluoroacetic acid (TFA). Samples were loaded onto the cartridge and washed 3× with 1 ml 0.1% TFA and 1× with 1 ml 1% FA, then eluted with 2 × 600 µl 50% MeCN/0.1% FA. Samples were dried down by vacuum centrifugation, then reconstituted, and their concentrations were measured by BCA assay. 400 µg aliquots were made based on the peptide-level concentration for TMT labeling.

### TMT labeling of peptides

TMT labeling was performed as previously described (*Zecha et al., 2019*). Briefly, 400 µg of peptides per sample were resuspended in 50 mM HEPES pH 8.5 at a concentration of 5 mg/ml. Dried TMT 10-plex reagent (Thermo Fisher Scientific) was reconstituted at 20 µg/µl in 100% anhydrous MeCN and added to samples at a 1:1 TMT to peptide mass ratio. The reaction was incubated for 1 hr at 25°C while shaking and quenched with 5% hydroxylamine to a final concentration of 0.2% for 15 min at 25°C while shaking. The TMT-labeled samples were then combined, dried to completion by vacuum centrifugation, reconstituted in 1 ml 0.1% FA, and desalted with a 100 mg SepPak cartridge as described above.

### Basic reverse-phase (bRP) fractionation

TMT-labeled peptides were fractionated via offline bRP chromatography as previously described (*Mertins et al., 2018*). Chromatography was performed with a Zorbax 300 Extend-C18 column (4.6 × 250 mm, 3.5 µm, Agilent) on an Agilent 1100 high-pressure liquid chromatography (HPLC) system. Samples were reconstituted in 900 µl of bRP solvent A (5 mM ammonium formate, pH 10.0 in 2% vol/vol MeCN). Peptides were separated at a flow rate of 1 ml/min in a 96 min gradient with the following concentrations of solvent B (5 mM ammonium formate, pH 10.0 in 90% vol/vol MeCN) 16% B at 13 min, 40% B at 73 min, 44% B at 77 min, 60% B at 82 min, 60% B at 96 min. A total of 96 fractions were collected and concatenated nonsequentially into 24 fractions. A total of 5% of each of the 24 fractions was reserved for global proteome analysis. The remaining 95% of each fraction were concatenated into 13 fractions for metal affinity chromatography and phosphoproteome analysis.

### Metal affinity chromatography

Ni-NTA Superflow Agarose Beads (QIAGEN) were prepared for metal affinity enrichment by performing 3× washes in HPLC water, 1 × 30 min incubation in 100 mM EDTA, 3× washes in HPLC water, 1 × 30 min incubation in 10 mM FeCl$_3$ in water (Sigma), and 3× washes in HPLC water. The beads were centrifuged to remove supernatant and resuspended in 1:1:1 acetonitrile:methanol:0.01% acetic acid solvent prior to incubation with peptides. Dried peptides were resuspended in 80% MeCN/0.1% TFA at a concentration of 0.5 µg/µl and incubated with 10 µl of beads for 30 min at room temperature with gentle end-over-end mixing and then centrifuged briefly to remove the flowthrough. The beads with phosphopeptides bound were transferred on top of a stage tip containing 2× C18 (Empore) discs and washed 3× with 100 µl of 80% MeCN/0.1% TFA and 1× with 50 µl 1% FA by centrifugation. Peptides were eluted from the beads and bound to the C18 discs using 225 µl IMAC elution buffer (500 mM K$_2$HPO$_4$, pH 7). Stage-tip desalting was performed with 1 × 100 µl 1% FA in water and eluted with 50 µl of 50% ACN/0.1% FA in water. Peptides were dried by vacuum centrifugation.

## Liquid chromatography and mass spectrometry

Dried fractions were reconstituted in 3% MeCN/0.1% FA to an estimated peptide concentration of 1 μg/μl for global proteome or by adding 8 μl per fraction for phosphoproteome fractions. Peptides were analyzed via coupled nanoflow LC-MS/MS using a Proxeon Easy-nLC 1000 (Thermo Fisher Scientific) coupled to an Orbitrap Q-Exactive Plus Mass Spectrometer (Thermo Fisher Scientific). A sample load of 1 μg (global proteome) or half of the available sample (phosphoproteome) for each fraction was separated on a capillary column (360 × 75 μm, 50°C) containing an integrated emitter tip packed to a length of approximately 25 cm with ReproSil-Pur C18-AQ 1.9 μm beads (Dr. Maisch GmbH). Chromatography was performed with a 110 min gradient of solvent A (3% MeCN/0.1% FA) and solvent B (90% MeCN/0.1% FA). The gradient profile, described as min:% solvent B, was 0:2, 1:6, 85:30, 94:60, 95:90, 100:90, 101:50, 110:50. Ion acquisition was performed in data-dependent acquisition mode with the following relevant parameters: MS1 orbitrap acquisition (70,000 resolution, 3E6 AGC target, 5 ms max injection time) and MS2 orbitrap acquisition (top 12, 1.6 $m/z$ isolation window, 30% HCD collision energy, 35,000 resolution, 5E4 AGC target, 120 ms max injection time, 2.1E4 intensity threshold, 20 s dynamic exclusion). The original mass spectra and the protein sequence databases used for searches have been deposited in the public proteomics repository MassIVE (http://massive.ucsd.edu) and are accessible at ftp://massive.ucsd.edu/MSV000087926/.

## MS data processing

MS/MS data were analyzed using Spectrum Mill v.7.00.208 (Agilent Technologies). MS2 spectra were extracted from RAW files and merged if originating from the same precursor, or within a retention time window of ±60 s and $m/z$ range of ±1.4, followed by filtering for precursor mass range of 750–6000 Da and sequence tag length >0. MS/MS search was performed against the mouse UniProt protein database downloaded on April 2021 and common contaminants, with digestion enzyme conditions set to 'Trypsin allow P,' <5 missed cleavages, fixed modifications (cysteine carbamidomethylation and TMT10 on N-term and internal lysine), and variable modifications (oxidized methionine, acetylation of the protein N-terminus, pyroglutamic acid on N-term Q, and pyro carbamidomethyl on N-term C). For phosphoproteome analysis, phosphorylation of S, T, and Y was added to the variable modifications. Matching criteria included a 30% minimum matched peak intensity and a precursor and product mass tolerance of ±20 ppm. Peptide-level matches were validated at a 0.8% FDR threshold and within a precursor charge range of 2–6. A second round of validation was then performed for protein-level matches, requiring a minimum protein score of 13 for the global proteome dataset. TMT10 reporter ion intensities were corrected for isotopic impurities in the Spectrum Mill protein/peptide summary module using the afRICA correction method that implements determinant calculations according to Cramer's Rule (*Shadforth et al., 2005*) and correction factors obtained from the reagent manufacturer's certificate of analysis for lot number SE240163. For global proteome analysis, protein-centric data, including TMT intensity values divided by the corresponding replicate control, were summarized in a table, which was further filtered to remove nonmouse contaminants and proteins with less than two unique peptides. For phosphoproteome analysis, peptide-centric data, including TMT intensity values divided by the corresponding replicate control, were summarized and filtered to remove nonmouse peptides.

## Immunoprecipitation and Western blotting

Shank3 protein was enriched by immunoprecipitation before analysis by Western blotting. At the end of scaling induction, neurons were lysed in RIPA buffer (150 mM NaCl, 50 mM Tris, 1% Triton X-100, 0.1% SDS, 0.5% sodium deoxycholate, 1 mM EDTA) containing cocktails of protease inhibitors (cOmplete, Sigma-Aldrich) and phosphatase inhibitors (PhosSTOP, Sigma-Aldrich), incubated on a rotating rocker at 4°C for 20 min, and centrifuged at 13,000 rpm for 15 min. The supernatants were then collected, and the protein concentration was measured using a commercial BCA assay (Thermo Fisher Scientific). To enrich the Shank3 protein, cell lysates (~800 ng) were incubated with guinea pig anti-Shank3 antibodies (1 μg, 162304, Synaptic Systems) on the rotating rocker at 4°C overnight. On the next day, the protein-antibody mixtures were incubated with magnetic protein-G beads (20 μl, Thermo Fisher Scientific) for another hour. After being washed thoroughly with RIPA buffer, the protein-antibody-bead mixtures were resuspended directly in the SDS-containing loading buffer (30 μl, LI-COR) and the Shank3 proteins were eluted into the buffer by heating at 70°C for

10 min. Once cooled down on the ice, the eluates were loaded into the 7% NuPAGE tris-acetate gel (Thermo Fisher Scientific), electrophoresed until well separated, and then slowly transferred to a PVDF membrane at 4°C overnight. Afterward, the membranes were incubated with the Odyssey blocking buffer (LI-COR) at room temperature for 1 hr and probed for Shank3 phosphorylation with a rabbit anti-pS1615 antibody (1:1000, Broad Institute) at 4°C overnight, followed by 1 hr incubation with the donkey anti-rabbit Alexa Fluor 680 antibody (1:5000, Jackson ImmunoResearch) at room temperature. After thorough washes, the membranes were imaged on a GelDoc imager (Bio-Rad). To measure total Shank3, the membranes were subject to a second round of staining where the guinea pig anti-Shank3 antibody (1:1000) and a donkey anti-guinea pig IRDye 800CW antibody were used. All the bands were visualized and quantified using the Image Lab Software (Bio-Rad). Data were collected from at least four independent experiments, in which all conditions were run in parallel with and normalized to the untreated control.

## Quantitative LC-MS/MS analysis of rat Shank3 phosphorylation

### Sample preparation
The immunoprecipitation method described above was used to extract the Shank3 protein for mass spectrometry with the following modifications: neurons from two 10 cm plates were pooled for each replicate. Two replicates were prepared for each condition (untreated and TTX-treated) and were subject to the electrophoresis protocol described above. Once separated, the bands containing Shank3 were visualized on the gel with the Coomassie Blue SafeStain (Thermo Fisher Scientific), cut out with a clean blade, and sent to Taplin Mass Spectrometry Facility at Harvard for further processing and analysis.

### In-gel digestion
Excised gel bands were cut into approximately 1 mm$^3$ pieces. The samples were reduced with 1 mM DTT for 30 min at 60°C and then alkylated with 5 mM iodoacetamide for 15 min in the dark at room temperature. Gel pieces were then subjected to a modified in-gel trypsin digestion procedure (*Shevchenko et al., 1996*). Gel pieces were washed and dehydrated with acetonitrile for 10 min, followed by removal of acetonitrile. Pieces were then completely dried in a speed-vac. Rehydration of the gel pieces was with 50 mM ammonium bicarbonate solution containing 12.5 ng/µl modified sequencing-grade trypsin (Promega, Madison, WI) at 4°C. Samples were then placed in a 37°C room overnight. Peptides were later extracted by removing the ammonium bicarbonate solution, followed by one wash with a solution containing 50% acetonitrile and 1% FA. The extracts were then dried in a speed-vac (~1 hr).

### TMT labeling
Samples were resuspended in 20 µl 200 mM HEPES buffer along with 6 µl of acetonitrile; 2 µl of TMT0 or TMT Super Heavy were added as light and heavy labels to each set of samples. After 1 hr, 2 µl of a 5% hydroxylamine was added for 15 min followed by the addition of 10 µl of 20% FA. Samples were mixed and then dried. Desalting of the samples was performed with an in-house de-salting tube using reverse phase C18 Empore SPE Disks (3M, Eagan, MN).

### Mass spectrometry analysis
On the day of analysis, the samples were reconstituted in 10 µl of HPLC solvent A (2.5% acetonitrile, 0.1% FA). A nano-scale reverse-phase HPLC capillary column was created by packing 2.6 µm C18 spherical silica beads into a fused silica capillary (100 µm inner diameter x ~ 30 cm length) with a flame-drawn tip (*Peng and Gygi, 2001*). After equilibrating the column, each sample was loaded via a Famos auto sampler (LC Packings, San Francisco, CA) onto the column. A gradient was formed and peptides were eluted with increasing concentrations of solvent B (97.5% acetonitrile, 0.1% FA). As each peptide was eluted, they were subjected to electrospray ionization, and then they entered into an LTQ Orbitrap Velos Pro ion-trap mass spectrometer (Thermo Fisher Scientific, San Jose, CA). Eluting peptides were detected, isolated, and fragmented to produce a tandem mass spectrum of specific fragment ions for each peptide. Peptide sequences (and hence protein identity) were determined by matching protein or translated nucleotide databases with the acquired fragmentation pattern by the software program, Sequest (Thermo Finnigan, San Jose, CA) (*Eng et al.,*

*1994*). The static modifications of 224.1525 mass units were set for lysine and the N-terminal of peptides (light label), along with 57.0215 mass units on cysteine (iodoacetamide). Differential modifications of 11.0243 mass units were set for lysine, and the N-terminal of peptides (heavy label), along with 79.9663 mass units to serine, threonine, and tyrosine, was included in the database searches to determine phosphopeptides. Phosphorylation assignments were determined by the Ascore algorithm (*Beausoleil et al., 2006*). All databases include a reversed version of all the sequences, and the data were filtered to between a 1 and 2% peptide FDR.

## Expression constructs and generation of Shank3 mutants
The construct expressing wild-type rat Shank3 (AJ133120.1) with an N-terminal GFP tag was obtained from Chiara Verpelli (*Verpelli et al., 2011*). Constructs expressing GFP-tagged or HA-tagged Shank3 phospho-mutants were generated using the Gibson Assembly kit (NEB) with the wild-type Shank3 as the template. For the double phospho-mimetic mutant, residues 1586 (TCC) and 1615 (AGC) were mutated from serine to aspartic acid (GAC). For the double phospho-deficient mutant, the same residues were mutated from serine to alanine (GCA). The coding regions of all constructs were fully sequenced to ensure that no unwanted random mutations were generated during cloning.

## Generation and validation of phosphospecific Shank3 antibody
Polyclonal antibodies targeting pS1615 Shank3 were generated in rabbits using the immunogen peptide AARLFS[pS]LGELSTI and purified against the phosphopeptide and counter selected against the non-phospho peptide using affinity chromatography (21st Century Biochemicals, Marlborough, MA). We then conducted two experiments to validate the specificity of the antibody. In the first experiment, we transfected HEK293T cells with either HA-tagged wild-type Shank3 or the S1586A/ S1615A mutant, and 48 hr later performed immunoprecipitation and immunoblotting as described above (*Figure 2—figure supplement 1A*). In the second experiment, protein lysates containing wild-type HA-Shank3 were subjected to the same Western blotting protocol except that after being transferred to the PVDF membrane, one set of the replicates were treated with lambda phosphatase (NEB) overnight at 25°C before incubation with the pS1615 antibody (*Figure 2—figure supplement 1B*).

## Identification of kinases that phosphorylate Shank3
Dissociated rat cortical neurons were prepared as described above. Cells were treated with CAMKII inhibitors KN62 (10 µM), KN93 (10 µM), or PKA inhibitor H89 (10 µM) for 40 min. During the last 10 min, PTX (100 µM) was added to half of the replicates to induce neuronal activities. Same volumes of DMSO were added to the control cells throughout the treatments. Afterward, cells were subject to the immunoprecipitation and Western blotting described above.

## PP2A activity assay
Dissociated cortical neurons were prepared as described above. After being treated with TTX for 1 hr or 24 hr, cells were lysed in phosphatase extraction buffer 20 mM imidazole-HCl, 2 mM EDTA, 2 mM EGTA, pH 7.0 with protease inhibitor cocktail (Roche cOmplete, MilliporeSigma, 589297001). Untreated sister cultures were prepared in parallel and served as controls. After sonication and centrifugation, protein concentrations of the supernatants were determined using the BCA assay and diluted to 500 µg/ml per sample. The same amounts of cell lysates were then subject to a commercial colorimetric PP2A activity assay (Millipore). In short, the chemically synthesized threonine phosphopeptide (denoted as pT in K-R-pT-I-R-R) was incubated with the endogenous PP2A pulled down from the cellular lysates. The degree of dephosphorylation is measured by the absorbance values at 650 nm with the addition of the Malachite Green Phosphate Detection Solution.

## Immunocytochemistry
24 or 48 hr after transfection, neurons were fixed with 4% paraformaldehyde for 15 min and subject to the established staining protocol (*Gainey et al., 2015*; *Tatavarty et al., 2020*). To probe endogenous or exogenously expressed Shank3, cells were permeabilized with the blocking buffer (0.1% Triton X-100/10% goat serum in PBS) at room temperature for 45 min. They were then incubated with the dilution buffer (5% goat serum in PBS) containing the following primary antibodies at 4°C overnight: chicken anti-GFP (1:1000, Aves Labs), guinea pig anti-Shank3 (1:1000, Synaptic Systems),

guinea pig or rabbit anti-VGluT1 (1:1000, Synaptic Systems). To stain surface GluA2, the protocol was modified such that prior to permeabilization cells were first incubated with mouse anti-GluA2 (1:1000, gift from Eric Gouaux, Vollum Institute, Oregon Health & Science University, Portland, OR) diluted in the blocking buffer without Triton X-100 for 1 hr at room temperature. On the next day, neurons were washed three times with PBS and incubated with the dilution buffer containing the following secondary antibodies at room temperature for 1 hr: goat anti-chicken Alexa-488, goat anti-mouse Alexa-555, goat anti-guinea pig Alexa-555, goat anti-guinea pig Alexa-647, and goat anti-rabbit Alexa-647 (1:400, Thermo Fisher Scientific). After three more washes with PBS, the glass bottoms containing stained neurons were detached from the dishes, mounted onto the slides using DAPI-Fluoromount-G mounting medium (SouthernBiotech), and sealed with nail polish.

## Image acquisition and analysis

All the images were acquired using a ×63 oil immersion objective on a laser-scanning confocal microscope (LSM880, Zeiss) using ZEN Black acquisition software. For all experiments, acquisition settings, including laser power, gain/offset, and pinhole size, were kept consistent. During image acquisition, pyramidal neurons were identified by their typical teardrop-shaped somata and apical-like dendrite. For each neuron, ~12 stacked images (step size: 0.33 μm) were obtained to include the apical dendrites and their dendritic branches, and then subject to maximum intensity projection using ZEN Black. To quantify the colocalization of synaptic proteins and their signal intensities, images were analyzed using the MetaMorph software (Molecular Devices). In all cases, GFP expression was first thresholded to create a mask that outlined the neuron. A region of interest was then manually drawn to include the dendritic branches distal to the primary branch point of apical-like dendrite. A threshold was set for the signal intensity in each channel to exclude background noise and was kept consistent across experimental groups. The granularity function in MetaMorph was then used to threshold puncta in each channel (puncta size: 0.5–5 μm). Binary images were generated to outline identified puncta in each channel, and the colocalized puncta were determined using the Logical AND operation. A synapse was defined as a punctum double-labeled with GFP and VGluT1 (*Figures 3D and 5*, *Figure 5—figure supplement 1*) or triple-labeled with GFP, sGluA2, and VGluT1 (*Figure 7*, *Figure 6—figure supplement 2*). Total puncta intensity for each channel at colocalized sites was then measured. The experimental conditions were always run in parallel with the control condition on sister cultures from the same dissociation, and the total puncta intensity measured in the experimental conditions was normalized to the mean total puncta intensity of control in sister cultures unless described otherwise.

## mEPSC recordings

Recordings were performed in whole-cell voltage clamp at room temperature; holding potential was –70 mV. Neurons with pyramidal morphology were targeted by visual inspection. Bath solution was ACSF containing (in mM) 126 NaCl, 5.5 KCl, 2 MgSO$_4$, 1 NaH$_2$PO$_4$, 25 NaHCO$_3$, 2 CaCl$_2$, 14 dextrose; and 25 μM PTX, 25 μM D-amino-5-phosphovaleric acid (AP5), and 0.1 μM TTX to isolate AMPA-mediated mEPSCs. Internal solution composition (in mM) was 120 KMeSO$_4$, 10 KCl, 2 MgSO$_4$, 10 K-HEPES, 0.5 EGTA, 3 K$_2$ATP, 0.3 NaGTP, and 10 Na$_2$ phosphocreatine. Dextrose was added to adjust osmolarity to 320–330 mOsm. Neurons were excluded if resting membrane potential (Vm) was >−55 mV, series resistance (Rs) was >20 MΩ, input resistance (Rin) was <100 MΩ, Rin or Vm changed by >30%, or <25 mEPSCs were obtained. mEPSCs were detected and analyzed using in-house software (see *Torrado Pacheco et al., 2021*); detection criteria included amplitudes >5 pA and rise times < 3 ms. To construct cumulative histograms, the first 25 events for each neuron were included.

## Statistical analysis

### Western blotting, PP2A activity assay, imaging, and electrophysiology experiments

GraphPad Prism software was used to conduct statistical analyses. For each experiment, data distribution in individual condition was tested for normality using the Anderson–Darling test. If all experimental conditions passed the normality test, a *t*-test, paired *t*-test, or two-way ANOVA was used where appropriate. If one or more conditions failed the normality test, a Mann–Whitney test or Kruskal–Wallis test was used as indicated. Significant Kruskal–Wallis tests were then subject to a Dunn's

post-hoc test for multiple comparisons. The significance levels are marked by asterisks (*): *p<0.05; **p<0.01; ***p<0.001; ****p<0.0001.

## Differential abundance analysis of proteomics data

Statistical analysis was performed in the R environment for statistical computing. Sample log2 TMT ratios were median-MAD normalized. Proteins and phosphosites with more than 50% missing values were removed. In order to identify proteins and phosphopeptides that respond to TTX or BIC treatment, a linear model with time groups as factors was fitted and a moderated *F*-test was performed on all coefficients using the limma package (*Ritchie et al., 2015*). Multiple hypothesis testing correction was performed using the Benjamini–Hochberg method. Bidirectionally regulated phosphosites across TTX and BIC treatment were those that showed significant regulation by *F*-test in both treatments, as well as opposite signs when calculating the mean logFC across all time points for each treatment.

## Pathway enrichment and network visualization

Proteins and phosphosites showing differential abundance in response to TTX or BIC treatment were clustered using the k-means methods (k = 4 for BIC and k = 3 for TTX). The optimal number of clusters was calculated using the elbow method using the total within sums of squares. Pathway enrichment analysis was performed for each cluster with the g:profiler tool (*Raudvere et al., 2019*). The background list of proteins was set to all detected in the proteomics analysis, and the databases used for annotation were Gene Ontology, KEGG, Wikipathways, and Reactome. The list of enriched pathways and genes contained in each pathway were exported to Cytoscape (*Shannon et al., 2003*). The EnrichmentMap app was used to generate a network of enriched pathways with the following parameters (min pathway p-value=0.01; overlap index = 1) (*Merico et al., 2010*).

## Interactive visualization of proteomic, phosphoproteomic data, and derived pathway activity scores

All proteome and phosphoproteome data can be explored as interactive R Markdown documents at https://proteomics.broadapps.org/HSP_TTX/ and https://proteomics.broadapps.org/HSP_Bic/. In addition to TTX/Bic-induced temporal changes in protein and phosphosite level, the apps enable the user to explore pathway-level changes across time points. To project protein and phosphosite expression matrices onto MSigDB canonical pathways (c2.cp v7.4), single-sample Gene Set Enrichment Analysis (ssGSEA) was used. TMT ratios of phosphosites and proteins mapping to the same gene symbol were combined by median expression prior to ssGSEA. The resulting normalized enrichment scores (NES) can be interpreted as *pathway activity scores* and serve as input for the longitudinal analysis described above.

## Acknowledgements

We thank Taplin Mass Spectrometry Facility (Harvard) for their assistance in processing and analyzing the rat MS experiment and Lirong Wang for her technical support. This work was supported by the NIH grants R35 NS111562 (GGT), F32 HL154711 (PMJB), Simons Foundation Award 345485 (GGT), and Stanley Center for Psychiatric Research (JRC).

## Additional information

### Funding

| Funder | Grant reference number | Author |
| --- | --- | --- |
| National Institute of Neurological Disorders and Stroke | R35 NS111562 | Gina G Turrigiano |
| Simons Foundation Autism Research Initiative | 345485 | Gina G Turrigiano |

| Funder | Grant reference number | Author |
|---|---|---|
| National Heart, Lung, and Blood Institute | F32 HL154711 | Pierre M Jean Beltran |
| Stanley Center for Psychiatric Research, Broad Institute | | Jeffrey R Cottrell |

The funders had no role in study design, data collection and interpretation, or the decision to submit the work for publication.

## Author contributions

Chi-Hong Wu, Conceptualization, Data curation, Formal analysis, Investigation, Methodology, Validation, Visualization, Writing – original draft, Writing – review and editing; Vedakumar Tatavarty, Conceptualization, Formal analysis, Investigation, Methodology, Validation, Writing – review and editing; Pierre M Jean Beltran, Data curation, Formal analysis, Methodology, Software, Validation, Visualization, Writing – original draft, Writing – review and editing; Andrea A Guerrero, Formal analysis, Investigation, Validation; Hasmik Keshishian, Investigation, Methodology, Supervision, Validation, Writing – review and editing; Karsten Krug, Data curation, Formal analysis, Software, Writing – review and editing; Melanie A MacMullan, Li Li, Investigation; Steven A Carr, Resources, Supervision, Writing – review and editing; Jeffrey R Cottrell, Funding acquisition, Resources, Supervision, Writing – review and editing; Gina G Turrigiano, Conceptualization, Funding acquisition, Project administration, Resources, Supervision, Visualization, Writing – original draft, Writing – review and editing

## Author ORCIDs

Chi-Hong Wu ![ORCID] http://orcid.org/0000-0001-6391-0747
Andrea A Guerrero ![ORCID] http://orcid.org/0000-0002-5324-8232
Melanie A MacMullan ![ORCID] http://orcid.org/0000-0001-5296-5902
Gina G Turrigiano ![ORCID] http://orcid.org/0000-0002-4476-4059

## Ethics

All experimental procedures were performed according to NIH guidelines and were approved by the Broad Institute of MIT and Harvard IACUC (mouse cultures) or the Brandeis University IACUC (rat cultures, protocol # 21002).

## Decision letter and Author response

Decision letter https://doi.org/10.7554/eLife.74277.sa1
Author response https://doi.org/10.7554/eLife.74277.sa2

---

# Additional files

## Supplementary files

• Transparent reporting form

## Data availability

All data generated or analyzed during this study are included in the manuscript and supporting files. Source data for Figures 2-7 have been provided. The original mass spectra and the protein sequence databases used for searches have been deposited in the public proteomics repository MassIVE (http://massive.ucsd.edu) and are accessible at ftp://massive.ucsd.edu/MSV000087926/.

The following dataset was generated:

| Author(s) | Year | Dataset title | Dataset URL | Database and Identifier |
|---|---|---|---|---|
| Wu C, Tatavarty V, Jean-Beltran PM, Guerrero A, Keshishian H, Krug K, MacMullan M, de Arce KP, Carr SA, Cottrell J, Turrigiano GG | 2021 | A Bidirectional Switch in the Shank3 Phosphorylation State Is Necessary to Enable Synaptic Scaling Up and Down | https://massive.ucsd.edu/ProteoSAFe/dataset.jsp?accession=MSV000087926 | MassIVE, 10.25345/C5CG2H |

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
