## [Editor Report]

The authors survey phosphorylation sites in a large array of proteins to identify targets involved in homeostatic synaptic plasticity in rodent neurons. They identified SHANK3, a critical scaffold protein in the postsynapse, as a target and show that SHANK3 can be phosphorylated and dephosphorylated to regulate homeostatic synaptic plasticity in both directions.

---

## [Decision Letter]

**Decision letter after peer review:**

Thank you for submitting your article "A Bidirectional Switch in the Shank3 Phosphorylation State Biases Synapses toward Up or Down Scaling" for consideration by *eLife*. Your article has been reviewed by 3 peer reviewers, one of whom is a member of our Board of Reviewing Editors and the evaluation has been overseen by Gary Westbrook as the Senior Editor. The following individual involved in review of your submission has agreed to reveal their identity: Peter Wenner (Reviewer #3).

Essential revisions:

1.The relationship between immunofluorescence and mEPSC recordings should be clarified under all conditions. In particular, Shank3 overexpression experiments should be validated in comparison to experiments with wild type levels of Shank3 in response to TTX and BIC.

2. The current findings should be put into context of the earlier work by the authors as well as others regarding the role of transcription/translation in synaptic upscaling and downscaling. IsShank3 phosphorylation/dephosphorylation permissive or instructive? Does it occur upstream of transcription regulation or is it a consequence of transcription regulation? These issues need further clarity.

3. Do the authors have any further insight into the identity of the kinase that phosphorylates Shank3.

4. Please justify the identification of PP2A, possibly by additional molecular manipulation, or directly addressing the short comings of their pharmacological approach.

*Reviewer #1 (Recommendations for the authors):*

The authors integrate extensive data on a comprehensive analysis of a phosphor-proteome associated with homeostatic synaptic down- and up-scaling. They identify 424 bidirectionally regulated phosphorylation sites and focus on the potential role of SHANK3, a key postsynaptic scaffold protein, as a primary causal target of bidirectional phosphorylation. The authors' experiments show that SHANK3 phosphomimetic mutations prevent scaling up but not down, whereas phosphodeficient mutations prevent scaling down but not up, suggesting a single reversible process regulating synaptic scaling. While the manuscript as it stands presents a critical advance in the field, given the authors' extensive work and contributions to this field, it is important that these observations are experimentally linked to earlier work.

1. Is this phosphorylation/dephosphorylation event downstream or upstream of transcriptional/translational regulation? Several groups, including earlier work from the senior author, demonstrated that synaptic scaling requires neuronal gene transcription or new protein synthesis (quite often as an instructive signal). Therefore, it is important to show if SHANK3 phosphorylation is upstream or downstream of transcription/translation. Does transcription or translation block impair SHANK3 phosphorylation/dephosphorylation?

2. The authors refer to several possible kinases, it would be great to pinpoint the identity of the kinase involved. In this way, one can address whether activity regulation can bidirectionally regulate this kinase.

3. Earlier work from the same group had suggested a critical and cell-autonomous role for MeCP2 in synaptic scaling up (Blackman et al., 2012). Within the context of comment #1 above is there a causal link between MeCP2 function and SHANK3 phosphorylation?

*Reviewer #2 (Recommendations for the authors):*

1. It is curious that the studies were conducted in relatively "young" neurons – only 7-10 days old in rat. Please clarify level of maturity and synaptic connectivity in these cultures. It would be surprising if these young neurons had dendritic spines, and this might have a big impact on the data that would be obtained. For example, Ehlers (2003) found that Shank levels were dynamically modulated by a 24 hr TTX/BIC treatment. How do levels of Shanks change in these cultures as they age? Or with TTX/BIC treatment? In addition, it is likely worth commenting on the Comment on the heterogeneity or otherwise of cell (neuron) types in the cultures used.

2. In discussing Figure 1E, I am not sure that describing the changes in Shank3 phosphorylation in response to BIC as robust and bidirectionally regulated (lines 170/171) is accurate/appropriate because this increase is only transient. The hyper-phosphorylation at rat S1615 was also transient and only observed after 10 minutes but reversed within 1 hour of BIC treatment. Has the scaling-down plasticity already started after 10 minutes of BIC treatment? If not, it may not be appropriate to say that Shank3 is hyperphosphorylated during scaling down. A more nuanced description of the findings would be preferred.

3. The change of phosphorylation of rat Ser1615, but not rat1586, during homeostatic plasticity is conserved across in mouse neurons. It leads to the question of whether rat Ser1586 phosphorylation is functionally relevant. The authors should compare the effects of Ser1615 single mutation to Ser1615/Ser1586 double mutation.

4. Comments on the PP2A studies:

A) The parallel use of okadaic acid, fostreicin and tautomycetin to probe the relative importance of PP2A vs.PP1 is strong, and the studies appear to have been carefully done. However, it would be good to show some sort of positive control to confirm that TAUT did indeed inhibit PP1 under these conditions.

B) For the PP2A assay, it appears that the lysates were prepared in the presence of phosphatase inhibitors. The authors need to define exactly what phosphatase inhibitors were used here. Since some commonly used inhibitors are generally thought to be irreversible toward PP2A, it is not clear how any phosphatase activity was detected.

C) The PP2A assay is based on immunoprecipitation of the PP2A catalytic subunit. However, it is likely that these samples contain an array of additional proteins. What did the authors do to confirm that the assay is indeed specifically measuring PP2A activity as opposed to that of other phosphatases that might be co-isolated? In addition, the authors should determine whether the levels of the PP2A catalytic subunit, or the co-precipitated regulatory/scaffolding subunits, are altered by homeostatic plasticity.

5. In Figure 6, how do the changes in EPSC amplitude induced by TTX or BIC treatment compare with those observed in neurons that are not over-expressing any Shank3? Or following the knockdown of endogenous Shank3 suppression? What were the EPSC frequencies? Were the frequencies affected by TTX or BIC?

6. The data in Figure 7 are not very compelling. First, I believe that these data should be analyzed using a 2-way ANOVA, since there are two variable factors (DMSO/FST and WT/AA). Second, some would argue that N here is the number of batches of neurons (independent transfections), rather than the number of cells. Please define the number of independent batches of neuronal cultures that were analyzed to generate the specified number of cells. Even if tighter analyses of these data continue to support the interpretation, there are several additional questions that should be addressed.

A) What about the colocalized punctae intensity in cells expressing WT or AA shank3 without TTX treatment?

B) What about the number of punctae per µm of dendrite?

C) What about the number of sGluA2 punctae in neurons that are not over-expressing exogenous Shank3? The selected images suggest that the number of sGluA2 punctae is dramatically increased in neurons expressing Shank3-AA vs Shank3-WT. Is Shank3-WT decreasing the number of punctae, or is Shank3-AA increasing the number of punctae?

*Reviewer #3 (Recommendations for the authors):*

This is a well considered discussion and excellent and thourough set of experiments which recognize the complexity of the scaling signaling pathways. As discussed above I believe the study is very strong and of broad importance. My main concern is a lack of discussion on the topic of the importance of phosphorylation vs location. Do the authors believe that dephosphorylation drives Shank 3 into the synapse and this localization is necessary for scaling, the phosphorylation state alone, or both localization and phosphorylation state. Do any of your results speak to these issues? The AA Shank 3 would be expected to go into the synapse but does not (Figure 3E) and yet it still allows for TTX upscaling (so here dephosphorylation matters more than location). Further, the hyperphoshporylation associated with down scaling is only transient expressed but is associated with a long term reduction in synaptic Shank 3 – here localization (out of the synapse) appears to be key for allowing downscaling not phosphorylation state.

In addition, it is not clear what we would expect in terms of puncta density, which might be expected to be reflected in mEPSC frequency (if I understand density), but changes in mEPSC frequency are not generally seen in these cultures treated with TTX or bicuculline. In Figure 3 you show that AA or DD Shank 3 has no change in puncta density which is fine, but in figure 5 TTX treatment leads to changes in puncta density – does this show up in mEPSC frequency?

---

## [Author Response]

Essential revisions:1.The relationship between immunofluorescence and mEPSC recordings should be clarified under all conditions. In particular, Shank3 overexpression experiments should be validated in comparison to experiments with wild type levels of Shank3 in response to TTX and BIC.

This is an important set of points. First, WT Shank3 overexpression does not impact mEPSC amplitude; baseline values here (Figure 6C-H) are indistinguishable from our previously reported values (e.g., Tatavarty et al., 2020), and this is consistent with our previously published data showing that Shank3 knockout or knockdown does not affect baseline mEPSC amplitude (Tatavarty et al., 2020). Second, due to biological variation the effect size of TTX treatment varies somewhat across datasets, but again is very similar in the Shank3 WT overexpression condition (~133% of control) and in empty-vector expressing neurons (~140% of control). The same is true when using GluA2 intensity as a measure of synaptic scaling; overexpression of Shank3 WT did not impact sGluA intensity at baseline, and TTX induced a similar increase in both conditions (Author response image 1). The effect size for the BIC/PTX condition is generally 10-20%, (Figure 6G, Figure 6 supplement 1), and is similar in the WT and control conditions. We now mention the lack of baseline effects in the Results section (lines 326-327).

**Author response image 1. sa2fig1:** Overexpressing wild-type Shank3 does not impact synaptic surface GluA2 intensity at baseline, or the magnitude of scaling up. Quantification of average synaptic sGluA3 intensity in neurons overexpressing Shank3-WT or a control GFP vector with and without TTX treatment (EV UN, n = 40, EV TTX, n = 45, WT UN, n = 33, WT TTX, n = 34). EV, empty vector; WT, Shank3-WT; UN, untreated.

2. The current findings should be put into context of the earlier work by the authors as well as others regarding the role of transcription/translation in synaptic upscaling and downscaling. IsShank3 phosphorylation/dephosphorylation permissive or instructive? Does it occur upstream of transcription regulation or is it a consequence of transcription regulation? These issues need further clarity.

Our data are consistent with a permissive role of Shank3 phosphorylation in gating upward and downward synaptic scaling, a point we now emphasize in the revised manuscript (abstract lines 39-41, lines 525-530). We did attempt to determine whether activity-dependent changes in Shank3 phosphorylation rely on transcription. We co-treated neuronal cultures with TTX and actinomycin D (ActD) for six hours; while we still saw a reduction in Shank3 phosphorylation during TTX treatment in the presence of ActD, the baseline effects of ActD were quite variable, so the experiment was inconclusive despite multiple replicates. While this is an interesting question, we feel it is not central to the main thrust of the manuscript, and note that it will require significant further study to determine how transcription-dependent and independent pathways interact to enable synaptic scaling.

3. Do the authors have any further insight into the identity of the kinase that phosphorylates Shank3.

To address this question, we performed additional experiments using kinase inhibitors.

Given that Shank3 is rapidly hyperphosphorylated in an activity-dependent manner, we focused on PKA and CAMKII, both of which are known to phosphorylate Shank3 at other sites (Wang et al., 2020b, Perfitt et al., 2020a). We treated the neurons with either CAMKII inhibitors (KN62 or KN93) or a PKA inhibitor (H89) for 40 minutes, then added PTX during the last 10 minutes to increase activity. Neither PKA nor CAMKII inhibition affected baseline Shank3 phosphorylation. In contrast, inhibition of either kinase prevented the Shank3 hyperphosphorylation induced by PTX. This result has two interesting implications: first, both kinase pathways are necessary for the activity-dependent phosphorylation of S1615. This suggests that Shank3 phosphorylation lies at the intersection of several complex signaling pathways involving different kinases. Second, the fact that baseline Shank3 phosphorylation was not significantly affected by these inhibitors suggests that these kinases are not required to maintain baseline levels of Shank3 phosphorylation. These data have been included in the revised Figure 4A-C.

4. Please justify the identification of PP2A, possibly by additional molecular manipulation, or directly addressing the short comings of their pharmacological approach.

We agree that a molecular manipulation of PP2A activity (such as overexpression of endogenous inhibitors, Tanimukai et al., 2005) would be ideal as an independent verification of our pharmacological results. However, in our hands inhibition of PP2A for durations longer than the 1 hr used here greatly impacted neuronal morphology and health. Overexpression or knockdown experiments lack the necessary temporal control to avoid this confound. We note that two distinct PP2A inhibitors gave the same results, and the specificity is further supported by the observation that PP2A inhibition only impacted activity-dependent phosphorylation of Shank3. We now discuss the limitations of our pharmacological approach in the Discussion section (lines 485-494).

Reviewer #1 (Recommendations for the authors):The authors integrate extensive data on a comprehensive analysis of a phosphor-proteome associated with homeostatic synaptic down- and up-scaling. They identify 424 bidirectionally regulated phosphorylation sites and focus on the potential role of SHANK3, a key postsynaptic scaffold protein, as a primary causal target of bidirectional phosphorylation. The authors' experiments show that SHANK3 phosphomimetic mutations prevent scaling up but not down, whereas phosphodeficient mutations prevent scaling down but not up, suggesting a single reversible process regulating synaptic scaling. While the manuscript as it stands presents a critical advance in the field, given the authors' extensive work and contributions to this field, it is important that these observations are experimentally linked to earlier work.1. Is this phosphorylation/dephosphorylation event downstream or upstream of transcriptional/translational regulation? Several groups, including earlier work from the senior author, demonstrated that synaptic scaling requires neuronal gene transcription or new protein synthesis (quite often as an instructive signal). Therefore, it is important to show if SHANK3 phosphorylation is upstream or downstream of transcription/translation. Does transcription or translation block impair SHANK3 phosphorylation/dephosphorylation?

This comment is addressed in Response to Essential revision 2.

2. The authors refer to several possible kinases, it would be great to pinpoint the identity of the kinase involved. In this way, one can address whether activity regulation can bidirectionally regulate this kinase.

This comment is addressed in Response to Essential revision 3.

3. Earlier work from the same group had suggested a critical and cell-autonomous role for MeCP2 in synaptic scaling up (Blackman et al., 2012). Within the context of comment #1 above is there a causal link between MeCP2 function and SHANK3 phosphorylation?

This is an interesting question, but would require an extensive series of additional experiments to explore any such link, and so is outside the scope of the current manuscript. We note that many pathways and molecular effectors have been implicated in synaptic scaling, and as a field we do not yet understand how all of these essential factors are coordinated to generate scaling up and down.

Reviewer #2 (Recommendations for the authors):1. It is curious that the studies were conducted in relatively "young" neurons – only 7-10 days old in rat. Please clarify level of maturity and synaptic connectivity in these cultures. It would be surprising if these young neurons had dendritic spines, and this might have a big impact on the data that would be obtained. For example, Ehlers (2003) found that Shank levels were dynamically modulated by a 24 hr TTX/BIC treatment. How do levels of Shanks change in these cultures as they age? Or with TTX/BIC treatment? In addition, it is likely worth commenting on the Comment on the heterogeneity or otherwise of cell (neuron) types in the cultures used.

First, we would like to clarify that most work on the mechanisms of synaptic scaling has been performed on relatively young neurons; the ages used here (cultures made from postnatal animals and used after 7-10 days in vitro) are identical to prior work from the Turrigiano lab. These young neurons have both spine and shaft excitatory synapses, and synaptic scaling is expressed at both types of synapses. We have exhaustively characterized synaptic, intrinsic, and network activity in these cultures over ~25 years of work; they exhibit robust network activity, and contain both excitatory and inhibitory neurons. We can visually target excitatory pyramidal neurons, the cell type we (and the field) perform the great majority of synaptic scaling work on. Finally, we note that synaptic scaling remains dependent on Shank3 in older mice (P28), as we showed recently using an in vivo induction paradigm (Wen and Turrigiano, 2021).

There are two key experimental differences between our and Ehlers’ studies (Ehlers, 2003): first, the neurons they used were cultured from embryonic mice, and the experiments were performed between 14-30 days in vitro; thus “age” (developmental time plus time in vitro) is slightly older but overlapping with our study. Second, they used longer (48 hours) pharmacological treatment. Despite these technical differences, Ehlers’ finding that synaptic Shank3 intensity (as well as the concentration of Shank3 in the PSD fraction) is bidirectionally regulated during TTX and BIC treatment is consistent with our results (Figure 3A), and consistent with our conclusion that synaptic Shank3 levels are dynamically modulated by activity.

2. In discussing Figure 1E, I am not sure that describing the changes in Shank3 phosphorylation in response to BIC as robust and bidirectionally regulated (lines 170/171) is accurate/appropriate because this increase is only transient. The hyper-phosphorylation at rat S1615 was also transient and only observed after 10 minutes but reversed within 1 hour of BIC treatment. Has the scaling-down plasticity already started after 10 minutes of BIC treatment? If not, it may not be appropriate to say that Shank3 is hyperphosphorylated during scaling down. A more nuanced description of the findings would be preferred.

We have rewritten this section of the manuscript to further emphasize that the hyperphosphorylation induced by BIC/PTX is transient, and describe in detail the differing time course of hyper and hypo phosphorylation and the implications (lines 202-212). Our data suggest that this transient hyperphosphorylation is necessary for initiating (but not maintaining) synaptic scaling down; this is based on several pieces of evidence, including that preventing this transient hyperphosphorylation with the AA mutant blocks scaling down, and this is now discussed in more detail (lines 459-466).

3. The change of phosphorylation of rat Ser1615, but not rat1586, during homeostatic plasticity is conserved across in mouse neurons. It leads to the question of whether rat Ser1586 phosphorylation is functionally relevant. The authors should compare the effects of Ser1615 single mutation to Ser1615/Ser1586 double mutation.

In an early pilot experiment, we tested whether mutations at 1615 were sufficient to block scaling; our pilot data suggested they were not, prompting us to mutate both sites, where we found robust effects on synaptic scaling. Because of the time-consuming nature of these experiments, we did not collect a full dataset on the single mutant and instead focused on the double mutant, but these pilot data do suggest that phosphorylation of both sites is important.

4. Comments on the PP2A studies:A) The parallel use of okadaic acid, fostreicin and tautomycetin to probe the relative importance of PP2A vs.PP1 is strong, and the studies appear to have been carefully done. However, it would be good to show some sort of positive control to confirm that TAUT did indeed inhibit PP1 under these conditions.

Unfortunately, we do not know of a robust assay for PP1 activity, so we were not able to perform this positive control. We agree the PP1 data are not as complete as the PP2A data, and it will await further studies to establish a clearer role for PP1 in controlling basal levels of Shank3 phosphorylation.

B) For the PPP2A assay, it appears that the lysates were prepared in the presence of phosphatase inhibitors. The authors need to define exactly what phosphatase inhibitors were used here. Since some commonly used inhibitors are generally thought to be irreversible toward PP2A, it is not clear how any phosphatase activity was detected.

We thank the reviewer for pointing out this error in our methods section; we doubled-checked our experimental procedures and confirmed that phosphatase inhibitors were NOT included in the buffers used in the PP2A assay. This error has been corrected in the revised methods section.

C) The PP2A assay is based on immunoprecipitation of the PP2A catalytic subunit. However, it is likely that these samples contain an array of additional proteins. What did the authors do to confirm that the assay is indeed specifically measuring PP2A activity as opposed to that of other phosphatases that might be co-isolated? In addition, the authors should determine whether the levels of the PP2A catalytic subunit, or the co-precipitated regulatory/scaffolding subunits, are altered by homeostatic plasticity.

First, we did not detect significant changes in expression or phosphorylation of PP2A catalytic, regulatory, or scaffolding subunits. We agree that we cannot rule out some degree of cross-reactivity in this assay, and differentiating between (e.g.) PP2A and PP1 relies on the same inhibitors that are also imperfectly specific. Despite these limitations, we feel that the multiple approaches we use taken together provide strong evidence for a specific role of PP2A. We have addressed these limitations in the Discussion section (lines 485-494).

5. In Figure 6, how do the changes in EPSC amplitude induced by TTX or BIC treatment compare with those observed in neurons that are not over-expressing any Shank3? Or following the knockdown of endogenous Shank3 suppression? What were the EPSC frequencies? Were the frequencies affected by TTX or BIC?

Regarding overexpression and synaptic scaling, see reply to essential revisions, point 1. We see no significant change in mEPSC frequency across conditions (Author response image 2); mEPSC frequency is highly variable and is not a reliable indicator of synaptic changes.

**Author response image 2. sa2fig2:** Overexpressing Shank3 phospho-mutants does not change mEPSC frequency at baseline or during the scaling protocols. Quantification of average mEPSC frequency in neurons pooled from the dataset in Figure 6 (WT, n = 16, WT + TTX, n = 9, WT + BIC, n = 8, DD, n = 7, DD + TTX, n = 11, AA, n = 9, AA + BIC, n = 14; Kruskal- Wallis test: p = 0.2483).

6. The data in Figure 7 are not very compelling. First, I believe that these data should be analyzed using a 2-way ANOVA, since there are two variable factors (DMSO/FST and WT/AA). Second, some would argue that N here is the number of batches of neurons (independent transfections), rather than the number of cells. Please define the number of independent batches of neuronal cultures that were analyzed to generate the specified number of cells. Even if tighter analyses of these data continue to support the interpretation, there are several additional questions that should be addressed.

This experiment was set up to explicitly test the difference between vehicle and FST for each condition, so we believe a Mann-Whitney test is appropriate for this analysis. Nonetheless a 2-way ANOVA also shows a significant interaction (p = 0.023 for sGluA2 intensity). Each data point represents a single transfected neuron, as is true for data throughout the manuscript. For each set of experiments, the data were pooled from 5 independent cultures. This (and most experiments in the manuscript) were performed after very low-efficiency transfection of cultures so that only a few pyramidal neurons were transfected per dish; in the past, we have treated the data in various ways, including performing nested statistics to account for variability across neurons, culture dishes, and dissociations, and have found that most variability arises from differences between neurons. Thus, we pool data across neurons and ensure that we include data from at least three independent dissociations for each experiment.

A) What about the colocalized punctae intensity in cells expressing WT or AA shank3 without TTX treatment?

The colocalized puncta intensity of Shank3 and sGluA in cells expressing Shank3 WT and AA without treatments are shown in Figure 3 (Shank3) and Figure 6 —figure supplement 2 (sGluA2), respectively. In summary, Shank3 AA overexpression did not significantly impact sGluA2 or Shank3 intensity compared to WT.

B) What about the number of punctae per µm of dendrite?C) What about the number of sGluA2 punctae in neurons that are not over-expressing exogenous Shank3? The selected images suggest that the number of sGluA2 punctae is dramatically increased in neurons expressing Shank3-AA vs Shank3-WT. Is Shank3-WT decreasing the number of punctae, or is Shank3-AA increasing the number of punctae?

We reanalyzed our data and quantified the synaptic density (puncta/µm) in each condition. We did not find an increase in the number of sGluA2-containing puncta in neurons overexpressing Shank3-AA. In the revised Figure 7D, we have included the quantification of puncta density (Figure 7D) and new images that better represent the mean results.

Reviewer #3 (Recommendations for the authors):This is a well considered discussion and excellent and thorough set of experiments which recognize the complexity of the scaling signaling pathways. As discussed above I believe the study is very strong and of broad importance. My main concern is a lack of discussion on the topic of the importance of phosphorylation vs location. Do the authors believe that dephosphorylation drives Shank 3 into the synapse and this localization is necessary for scaling, the phosphorylation state alone, or both localization and phosphorylation state. Do any of your results speak to these issues? The AA Shank 3 would be expected to go into the synapse but does not (Figure 3E) and yet it still allows for TTX upscaling (so here dephosphorylation matters more than location). Further, the hyperphoshporylation associated with down scaling is only transient expressed but is associated with a long term reduction in synaptic Shank 3 – here localization (out of the synapse) appears to be key for allowing downscaling not phosphorylation state.

The reviewer made an excellent point that there is more than one possible mechanism by which Shank3 phosphorylation could gate synaptic scaling. We agree that the Shank3 phosphorylation state could influence its localization, and/or mediate its interaction with cellular factors in local signaling pathways. These two mechanisms are not mutually exclusive, and their relative importance likely depends on the direction of scaling, the phosphorylation state, and the timing of phosphorylation. During scaling up, persistent dephosphorylation of Shank3 may locally stabilize Shank3 in the synapse and modulate its interaction with other synaptic proteins that result in GluA2 accumulation. During scaling down, the early but transient hyperphosphorylation may displace Shank3 and is possibly crucial for initiating molecular programs that then sustain scaling down. Future experiments to dissect differences in compartmentalized protein interactomes of Shank3 WT, DD, and AA during scaling up and down would shed light on these complex processes. The revised manuscript includes a discussion of these possibilities (lines 495-520).

In addition, it is not clear what we would expect in terms of puncta density, which might be expected to be reflected in mEPSC frequency (if I understand density), but changes in mEPSC frequency are not generally seen in these cultures treated with TTX or bicuculline. In Figure 3 you show that AA or DD Shank 3 has no change in puncta density which is fine, but in figure 5 TTX treatment leads to changes in puncta density – does this show up in mEPSC frequency?

We see no significant changes in mEPSC frequency across conditions Author response image 2. This is not surprising for two reasons. First, as mentioned above in response to reviewer 2, mEPSC frequency is quite variable due to many factors that are unrelated to functional synapse density, so the correlation is poor. Second, not all excitatory synapses express detectible levels of Shank3; an increase in Shank3 intensity can thus increase the colocalization rate between Shank3 and sGluA2 without signifying a change in the number of functional synapses.